# A V0 core neuronal circuit for inspiration

Jinjin Wu[1], Paolo Capelli[2,3], Julien Bouvier [1], Martyn Goulding[4], Silvia Arber[2,3] & Gilles Fortin [1]

Breathing in mammals relies on permanent rhythmic and bilaterally synchronized contractions of inspiratory pump muscles. These motor drives emerge from interactions between critical sets of brainstem neurons whose origins and synaptic ordered organization remain obscure. Here, we show, using a virus-based transsynaptic tracing strategy from the diaphragm muscle in the mouse, that the principal inspiratory premotor neurons share V0 identity with, and are connected by, neurons of the preBötzinger complex that paces inspiration. Deleting the commissural projections of V0s results in left-right desynchronized inspiratory motor commands in reduced brain preparations and breathing at birth. This work reveals the existence of a core inspiratory circuit in which V0 to V0 synapses enabling function of the rhythm generator also direct its output to secure bilaterally coordinated contractions of inspiratory effector muscles required for efficient breathing.

[1] Paris-Saclay Institute of Neuroscience, 1 Avenue de la Terrasse, 91190 Gif sur Yvette, France. [2] Biozentrum, Department of Cell Biology, University of Basel, 4056 Basel, Switzerland. [3] Friedrich Miescher Institute for Biomedical research, 4058 Basel, Switzerland. [4] Salk Institute for Biological Studies, 10010 North Torrey Pines Road, La Jolla, CA 92037, USA. Correspondence and requests for materials should be addressed to G.F. (email: gilles.fortin@cnrs.fr)

I n mammals, breathing is a motor behavior generated by a central pattern generator (CPG) located in the brainstem and spinal cord that produces rhythmic contraction of muscles that regulates lung volume and control upper airway patency to maintain bodily homeostasis[1]. The respiratory CPG in rodents is precociously active in the fetus at around two thirds through gestation[2, 3], allowing a period of breathing practice prior to the challenge of encountering air at birth. Starting at birth, the respiratory CPG continuously adapts the frequency and amplitude of the respiratory motor command to metabolic demands linked to exercise and environmental changes. Thus, the respiratory CPG regulates the choice, the timing and the intensity of activation of appropriate groups of premotor neurons, motor neurons, and their muscle targets. The CPG must probably do so respecting two constraints, namely the synchronicity and the balanced amplitude of the motor drives onto left and right respiratory effector muscles (e.g., left and right costal diaphragm muscles that are the prime movers of tidal air). The identity of neurons in charge of securing bilaterally synchronized and amplitude balanced inspiratory motor drive is investigated here.

Over the past decade, strategies exploiting the history of gene expression by neural progenitors or precursors have allowed the manipulation of neurons with unprecedented specificity to reveal their role in circuit function and behavior[4, 5]. In that way, we established that the preBötzinger complex (preBötC) that paces inspiration[6] is composed of interconnected rhythmogenic V0 type neurons (i.e., deriving from p0 progenitors expressing the transcription factor Dbx1), which are synchronized with their contralateral cognate neurons by commissural projections established through the roundabout homolog 3 (Robo3) signaling pathway[7]. Therefore, bilateral synchronicity of the respiratory motor command is at least in part built-in at the level of the rhythm generator.

Although inspiratory descending circuits have been described for adult rodents and cats[8–10], nothing is known of the origin of premotor neurons downstream of the rhythm generator that secure temporal and amplitude patterning of the inspiratory motor drive. Here, we addressed this question in early postnatal mice using monosynaptic viral-based circuit-mapping approaches that allow unambiguous identification of phrenic premotor neurons (Ph-preMNs)[11]. We find that Ph-preMNs are distributed at several sites of the brainstem and include individual neurons with bifurcating axons that connect to phrenic motor neurons (Ph-MNs) on both sides of the midline. The main premotor relay is the rostral ventral respiratory group (rVRG), abutting the pre-BötC caudally. These rVRG neurons gain prominence over the prenatal period and end up forming at birth, together with the preBötC, the core inspiratory circuit that generates the rhythm and secures bilaterally synchronous and balanced drives to Ph-MNs required for efficient breathing. Strikingly, rVRG and pre-BötC neurons, found both glutamatergic and harboring commissural axons, share a common origin in p0 progenitors, highlighting the centrality of Dbx1-expressing neural progenitors in the advent of aspiration breathing in vertebrates.

## Results

**Mapping phrenic premotor neurons in early postnatal mice.** To selectively label neurons that synapse onto Ph-MNs, we used transsynaptic rabies technology with monosynaptic restriction. This method makes use of a glycoprotein-G-deleted mutant rabies virus (ΔG-Rb) whose retrograde transsynaptic spread from infected source cells (here Ph-MNs), requires complementation in these cells by the rabies glycoprotein-G (G)[11, 12]. Once inside presynaptic neurons, the deficient virus ceases to spread for lack of G, and thus only phrenic premotor neurons are traced safe of

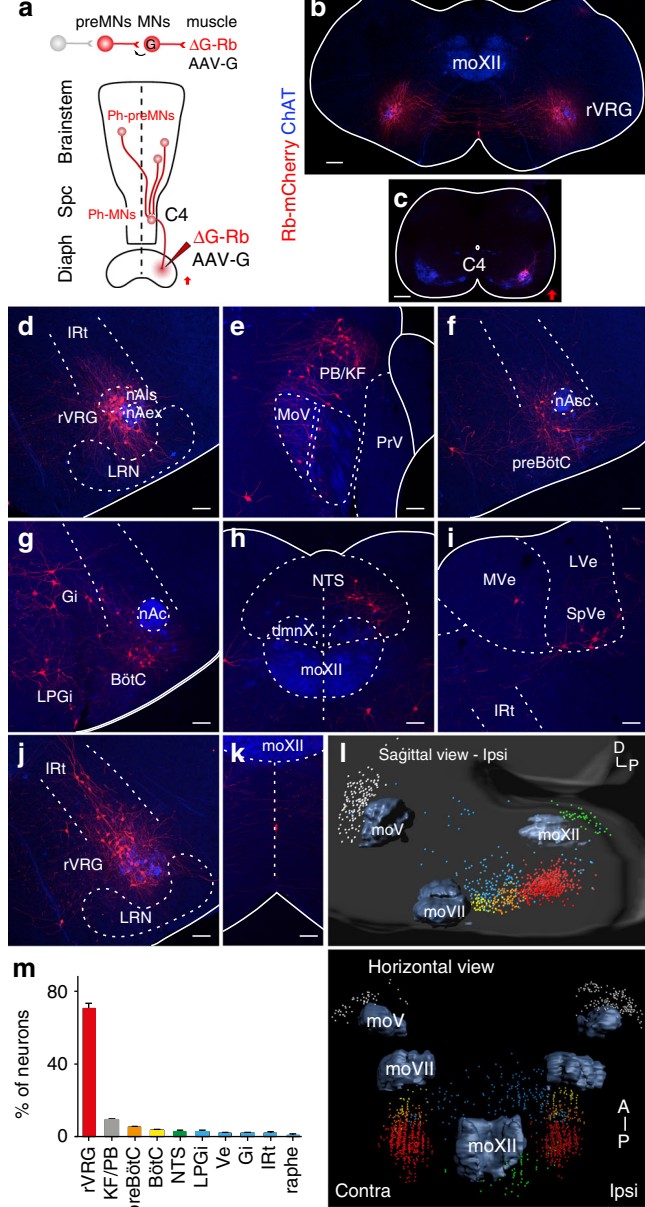

**Fig. 1** Distribution of Ph-preMNs in P9 mice following unilateral viral injections of the diaphragm. **a** Monosynaptic tracing scheme for Ph-preMNs using a G-deficient Rb virus (ΔG-Rb) and a G-coding adeno-associated (AAV-G) viral cocktail. **b**, **c** Brainstem and cervical spinal transverse sections showing, respectively, bilaterally distributed trace⁺ Ph-preMNs of the rVRG (**b**) and ipsilaterally located seeding trace⁺ Ph-MNs. **d**–**k** Representative images of transverse sections of the brainstem showing (in decreasing abundance order) trace⁺ Ph-preMNs in the rVRG (**d**), in the PB/KF (**e**), in the preBötC (**f**), in the BötC, Gi and LPGi (**g**), in the NTS (**h**), in Ve nuclei (**i**), in the IRt (**j**), and in the raphe (**k**). **l** 3D reconstructions of the brainstem showing the spatial distribution of Ph-preMNs (*n* = 4 cumulated counts) in sagittal (*top*) and horizontal (*bottom*) views (color code of locations is given in **m**). **m** Summary histogram of the distribution Ph-preMNs (percent of total trace⁺ cells). dmnX, dorsal motor nucleus of the vagus, LRN, lateral reticular nucleus; LVe, lateral vestibular nucleus; MoV, trigeminal motor nucleus; MoXII, hypoglossal motor nucleus; MVe, medial vestibular nucleus; nAc, compact nucleus ambiguus; nAex, external formation of nucleus ambiguus; nAls loose formation of nucleus ambiguus; nAsc, semi-compact nucleus ambiguus; PrV, principal sensory trigeminal nucleus; Spc, spinal cord; SpVe, spinal vestibular nucleus. *Scale bars* **b**, **c** 200 μm; **d**–**k** 100 μm

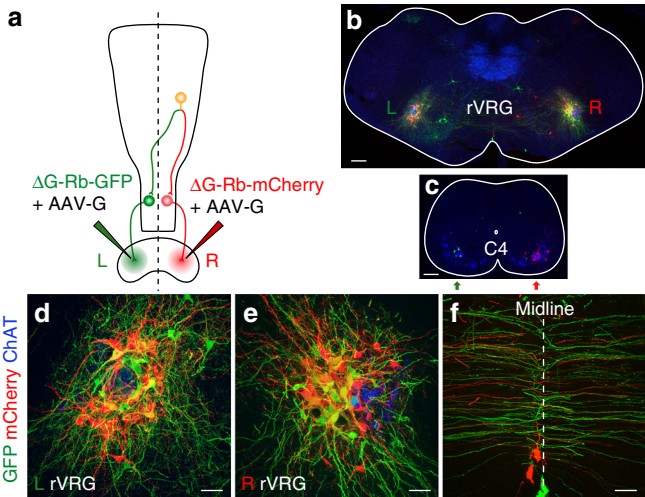

**Fig. 2** Individual Ph-preMNs in the rVRG project bilaterally on Ph-MNs. **a** Tracing scheme based on injections of a *green* virus (Rb-GFP) in the *left* diaphragm (L green lettering) and of a *red* virus (Rb-mCherry) in the *right* diaphragm (R red lettering). **b**, **c** Transverse sections at the level of the rVRG (**b**) and at the C4 level (**c**). Note the presence of double labeled (GFP$^+$/mCherry$^+$, yellow) rVRG neurons on the *left* and *right* side (**b**) while seeding Ph-MNs (**c**) on each side express exclusively either GFP (*green*) or mCherry (*red*). **d–f** Close-up view of the *left* (**d**) and *right* (**e**) rVRG showing exclusive *green* or *red* cells as well as double labeled cells (*yellow*). **f** Close-up view of labeled commissural axons over the midline (*dotted line*). Scale bars: **b**, **c** 200 μm; **d–f** 50 μm

the confounds normally associated to multi-synaptic jumps of non-deficient rabies virus. ΔG-Rb-mCherry and an adeno-associated virus (AAV) expressing G (AAV-G), were co-injected in the diaphragm of P1 mice ($n = 21$) to retrogradely infect Ph-MNs and initiate transsynaptic spread to their premotor partners (Fig. 1a). As expected, at P9, Ph-MNs labeled with Rb-mCherry (thereafter trace$^+$) were found in the ventral spinal cord over cervical segments C3-C6 exclusively on the injected side (Fig. 1c). Premotor neurons transsynaptically labeled with mCherry were observed bilaterally in the pons and medulla of the brainstem (Fig. 1b) and in cervical spinal segments, but in no other location. The large majority of Ph-preMNs were found in the rostral ventral respiratory group (rVRG, $399 \pm 163$ neurons or $70.6 \pm 2.7\%$ of all Ph-preMNs, Fig. 1d, m). Others were found in the following locations (Fig. 1m, in decreasing order of abundance; $n = 6$ mice): the parabrachial nuclei and the Kölliker-Fuse nucleus (PB/KF) in the dorsolateral pons (collectively $51 \pm 21$ neurons; $9.2 \pm 0.6\%$ Fig. 1e); the area of the preBötzinger complex (preBötC, $28 \pm 11$ neurons; $5.0 \pm 0.5\%$ Fig. 1f); the Bötzinger complex (BötC, $21 \pm 8$ neurons; $3.5 \pm 0.4\%$ Fig. 1g); the nucleus of the solitary tract (NTS, $18 \pm 7$ neurons; $2.7 \pm 0.9\%$ Fig. 1h). Other smaller premotor groups were also identified in the brainstem (see Fig. 1g–k) and we aligned serial sections to generate a three-dimensional (3D) reconstruction of all premotor neuronal populations (Fig. 1l; $n = 4$ mice). Ph-preMNs were not found outside of the brainstem except in the cervical spinal cord in small numbers in laminae VII and X throughout cervical segments C2-C6 (see Supplementary Fig. 3j). Interestingly, the unilateral tracing scheme revealed that the very large majority of Ph-preMNs were found equally distributed on both sides of the midline; only the PB/KF, the NTS and the Gi showed ipsilaterally biased projections (Supplementary Table 1). These observations were confirmed using an alternative method with centrally targeted motor neuron infection to initiate transsynaptic spread (Supplementary Fig. 1).

To directly visualize the bilateral descending projections of Ph-preMNs, we injected the left diaphragm with a ΔG-Rb-GFP (green) and the right diaphragm with a ΔG-Rb-mCherry (red) (Fig. 2a), both complemented with AAV-G. While, as expected, Ph-MNs exclusively expressed GFP or mCherry, on the left and right side, respectively (Fig. 2c), rVRG neurons and axons were found that co-expressed GFP and mCherry (yellow, Fig. 2b, d, e, f), demonstrating that they individually synapsed onto both, left and right side Ph-MNs (Fig. 2). Such doubly infected neurons were also present in other Ph-premotor areas, the PB/KF (Supplementary Fig. 2a, b), the NTS (Supplementary Fig. 2c), the BötC (Supplementary Fig. 2d) and the preBötC (Supplementary Fig. 2e). Altogether, these data demonstrate that the rVRG hosts neurons with a descending branched axon terminating on Ph-MNs on both sides of the midline (Supplementary Fig. 2f), the simplest cellular design securing synchronous and amplitude balanced motor drives to the left and right diaphragm muscles. As the rVRG outnumbered all other premotor stations by an order of magnitude and contributed to about 70% of all Ph-preMNs, we focused on it for further analysis.

**Origin and neurotransmitter phenotype of rVRG neurons.** With the intention of manipulating rVRG neurons genetically, we next sought to identify their origins as well as their excitatory or inhibitory nature. We performed viral injections for tracing of premotor neurons in diaphragm muscles in mouse lines in which a cre allele is driven by the promoter of the transcription factors Dbx1 (*Dbx1$^{creERT2}$*), Engrailed1 (*En1$^{cre}$*), Lbx1 (*Lbx1$^{cre}$*), Sim1 (*Sim1$^{cre}$*) or of the vesicular glutamate transporter 2 (*vGlut2$^{cre}$*), and vesicular GABA transporter (*vGAT$^{cre}$*) using a *Tau-nls-lacZ* allele as a Cre-dependent reporter. As illustrated in Fig. 3, in the rVRG at P9, about two thirds (range 59.3–77.8%, 65.5 ± 4.2%, $n = 4$) of trace$^+$ neurons expressed LacZ in a *Dbx1$^{creERT2}$* background (Fig. 3a), whereas 21.9 ± 1.8% (range 18.3–23.6%, $n = 3$) did so in a *Lbx1$^{cre}$* background (Fig. 3b) and none did in either the *En1$^{cre}$* or the *Sim1$^{cre}$* background (Supplementary Fig. 3a, b). In addition, we found that trace$^+$ rVRG neurons expressed neither *Phox2b* (a dB2 marker, Supplementary Fig. 3c) nor *Tlx3* (a dB3 marker, Supplementary Fig. 3d). These data indicate that the majority of rVRG neurons originated from Dbx1-expressing ventral progenitors and have thus a V0 identity, while the complement likely originates from dorsal dB1/dB4 progenitor domains marked by expression of *Lbx1* but lack of *Tlx3* and *Phox2b* expression[13].

We found that most (74.7 ± 2.9%, $n = 4$ mice) trace$^+$ rVRG neurons were excitatory (vGlut2$^+$), among which 90% expressed the transcription factor *Pax2* (Fig. 3c), while 17.2 ± 2.5% ($n = 4$) were inhibitory (vGAT$^+$), all of them lacking *Pax2* expression (Fig. 3d). This indicates that *Pax2* expression can be used as a proxy for an excitatory phenotype. *Pax2* was expressed in 95.8 ± 0.1% ($n = 3$) of trace- rVRG neurons with V0 identity (Fig. 3e), whereas *Pax2* expression was missing in 83.5 ± 2.9% ($n = 3$) of trace + rVRG neurons with dB identity (Fig. 3f). The inhibitory status of rVRG trace$^+$ neurons recapitulated through *Lbx1$^{cre}$* was confirmed by in situ hybridization to *GAD1* and *GlyT2* (Supplementary Fig. 3e, f), in agreement with the notion that dB1 and/or dB4 domains are the exclusive providers of inhibitory dB neurons[13]. Taken together, these findings imply that in the rVRG, neurons with a dB identity contribute the full complement of inhibitory neurons, while the vast majority of neurons are excitatory and have V0 identity.

We next questioned whether other Ph-preMN groups shared V0 identity with the rVRG. The Ph-preMNs located in the PB/KF nucleus (all vGlut2$^+$, data not shown) were fully recapitulated in *En1$^{cre}$* background (Supplementary Fig. 3g) as expected from

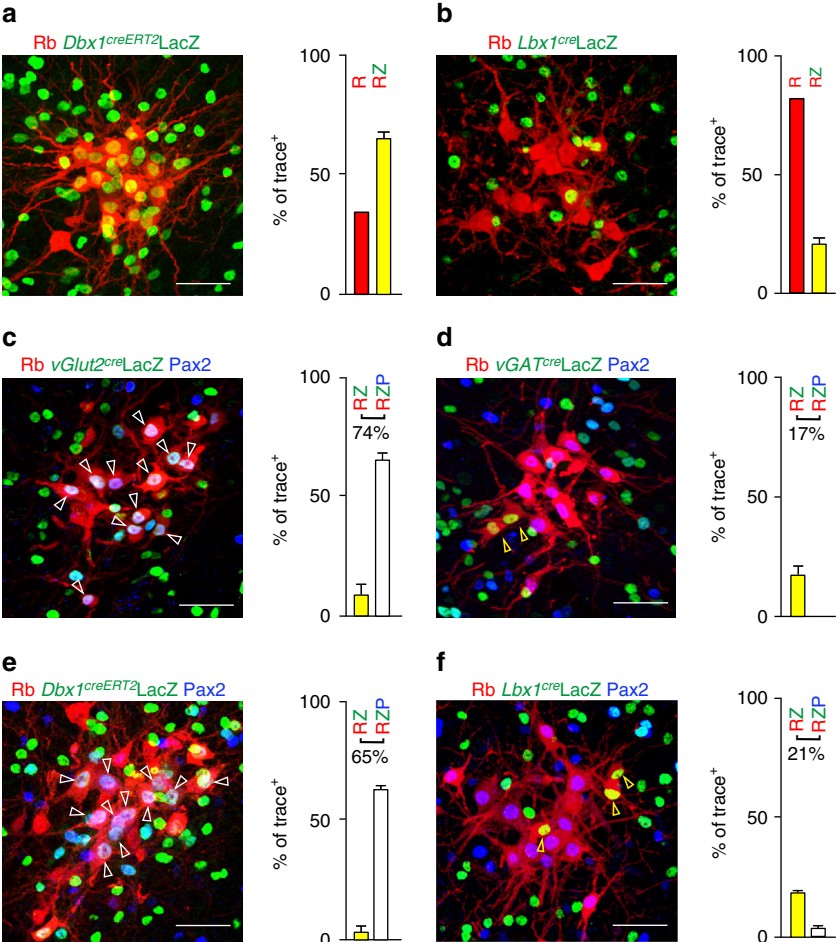

**Fig. 3** Identity of the Ph-preMNs of the rostral ventral respiratory group. **a, b** Transverse brainstem section showing trace[+] rVRG neurons labeled by Rb-mCherry (*red*, R) and counterstained for nuclear expression of LacZ (*green*, Z) and summary histograms featuring the percentage of trace[+] rVRG neurons expressing LacZ (*yellow bars*, RZ). The rVRG is comprised of neurons with a history of expression of Dbx1 (**a**) or Lbx1 (**b**). **c–f** Same as above with additional immunostaining for Pax2 (*blue*, P) and summary histograms showing the percentage of trace[+] rVRG neurons expressing LacZ alone (*yellow bars*, RZ, *yellow arrowhead*) or co-expressed with Pax2 (*white bars*, RZP, *white arrowhead*) when LacZ is expressed from the vGlut2 locus (**c**); from the vGAT locus (**d**); from the Dbx1 locus (**e**) and from the Lbx1 locus (**f**). Note the comparable proportion of triple positive cells in **c**, **e** panels, of double positive cells in **d**, **f** panels and the virtual absence of triple positive cells in **d**, **f** panels. *Scale bars*: 50 μm

their known rostral and dorsal rhombencephalic origin[14]. In the BötC area, Ph-preMNs (all vGAT[+], data not shown) were derived from Lbx1-expressing precursors (Supplementary Fig. 3h)[15]. The Ph-preMNs in the NTS (all vGlut2[+], data not shown) expressed *Phox2b* as expected from their origin in the dA3 domain of progenitors (Supplementary Fig. 3i)[16, 17]. The only exception in the brainstem was the fraction of trace[+] Ph-preMNs located in the preBötC area that also derived from Dbx1-expressing progenitors and were vGlut2[+] and Pax2[+] but NK1R[−] (Supplementary Fig. 4d–f). These premotor neurons cannot presently be distinguished from rVRG neurons (they could just represent their most rostral contingent) and will be referred to as preBötC Ph-preMNs, only to indicate their location. Occasionally, we noted the presence at spinal level of V0 partition-like cholinergic cells (Supplementary Fig. 3j). Therefore, these data establish that the trace[+] glutamatergic Ph-preMNs with V0 identity reside almost exclusively in the rVRG.

**Projections of rVRG neurons.** To determine whether excitatory rVRG neurons have other synaptic targets in addition to Ph-MNs, we exploited the intense mCherry labeling resulting from monosynaptic retrograde labeling in a *Dbx1creERT2; TausynGFP* mouse, where synaptic terminals formed by Dbx1-derived

neurons are strongly labeled with synGFP. In this context, the synaptic terminals of trace[+] rVRG neurons will be doubly labeled by mCherry and GFP and will be discriminated from those arising from other trace[+] premotor groups (mCherry[+] only) or V0 trace[−] neurons (GFP[+] only). We verified first the massive presence of trace[+] rVRG neuron synaptic terminals ($7.7 \pm 0.52$ boutons/soma, $n = 62$ cells) on motor neurons of the Ph-MN pool (Fig. 4a–c). This contrasted with cranial motor neuron pools that featured fibers and occasional boutons from Ph-preMNs (red), many boutons from trace[−] V0 neurons (green), but virtually no bouton from trace[+] rVRG neurons (mCherry[+]/GFP[+], yellow) (Supplementary Fig. 5). Apart from Ph-MNs, the only conspicuous presence of trace[+] rVRG synaptic terminals was in the ipsi- and contra-lateral lateral reticular nucleus (LRN), a pre-cerebellar hub structure that also receives ascending signals from multiple spinal premotor sources (Fig. 4g–i)[18]. In particular, we noted the absence of doubly labeled terminals on trace[+] rVRG neurons themselves ($0.07 \pm 0.04$ boutons/soma, $n = 54$ cells, Fig. 4d–f). Altogether these data indicate that trace[+] rVRG neurons are not intrinsically connected and that they send collateral projections to the LRN but not to major respiratory-related cranial motor neurons. In addition, we detected abundant GFP[+] only terminals on trace[+] rVRG neurons, indicating that they receive

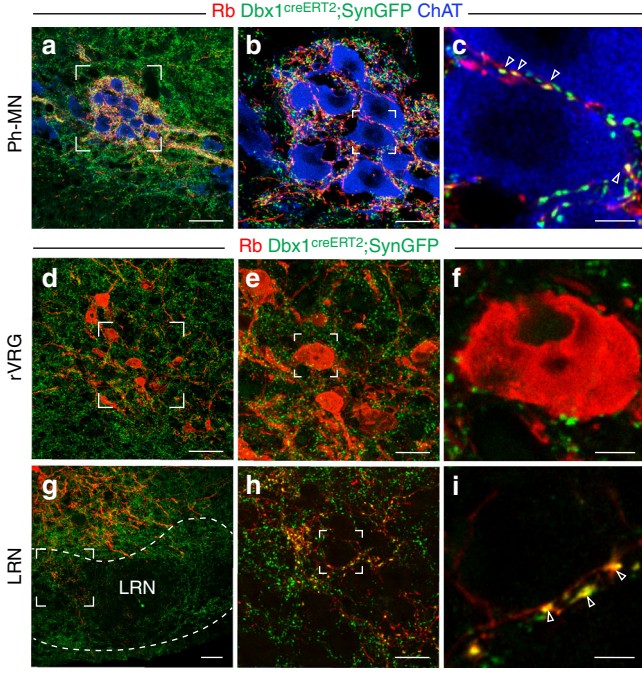

**Fig. 4** Synaptic targets of V0 rVRG neurons. **a** V0 rVRG synaptic terminals double labeled (*yellow*) by rabies-mCherry (Rb, *red*) and SynGFP (*green*) are massively present on Ph-MNs (trace⁻, ChAT⁺, *blue*) contralateral to the diaphragm viral injection side. **b** Zoom on the *square inset* in **a**. **c** Single optical section of the *inset* in **b** showing individual synapses (*arrowheads*). **d**–**f** rVRG (trace⁺, *red*) neurons are devoid of double labeled terminals. **e** Zoom of the *inset* in **d**. **f** Single optical section of the *inset* in **e**. **g** V0 trace⁺ rVRG neurons project to the lateral reticular nucleus (LRN). **h** Zoom of the inset in **g**. **i** Single optical section of the inset in **h** showing individual V0 rVRG synaptic terminals (*arrowheads*) presumably abutting the soma of a LRN neuron. *Scale bars*: **a**, **d**, **g** 50 μm; **b**, **e**, **h** 20 μm; **c**, **f**, **i** 5 μm

inputs from other V0-type neurons that we next set out to identify.

**PreBötC V0 neurons connect rVRG V0 neurons.** A candidate source of V0 inputs to the rVRG is the preBötC whose neurons form the inspiratory rhythm generator[7, 19]. To verify this, we modified the original viral tracing scheme to enable an additional retrograde synaptic jump of the ΔG-Rb from V0 neurons. Viral injections were made in Dbx1[creERT2]; R26[ssHTB] pups in which G-complementation and nuclear GFP labeling occurs in V0 neurons thanks to cre-dependent expression of, respectively, G- and histone 2B-GFP-encoding transgenes (Fig. 5a). In this background, the distribution and the number of trace⁺ neurons in Ph-preMN stations was as in monosynaptic traced pups, but an additional large neuronal population was visualized just anterior to and comparable in size with the rVRG, in the preBötC area (Fig. 5b, c). This neuronal cluster was largely composed of V0 neurons (Fig. 5d–g). These data demonstrate the presence of abundant homotypic V0-V0 synapses between presynaptic preBötC V0 neurons and postsynaptic rVRG V0 neurons.

**V0 neurons ensure efficient bilateral breathing drives.** We then assessed the contribution of synapses made by V0 neurons of the rVRG and preBötC to the mounting of the inspiratory motor drive. First, we tested the ability of rVRG V0 neurons to control the Ph-MNs outputs bilaterally during fetal breathing. At embryonic day (E) 15.5, in brainstem spinal cord preparations ($n = 4$), a unilateral injection of a rhodamine dextran dye in the

phrenic motor column led to bilateral back-labeling of rVRG neurons (Fig. 6a, b) revealing that bilateral descending projections of the rVRG neurons are already established at this stage. We then photo-stimulated the rVRG V0 neurons, in E15.5 Dbx1[creERT2]; ChR2-tdTomato preparations transversally cut so as to expose the rVRG and eliminate the preBötC (Fig. 6c). In such preparations ($n = 5$), the phrenic nerves, which exit through the fourth cervical roots (C4) are spontaneously silent. When light was targeted to the rVRG on one side, it activated the ipsi- but not the contra-lateral rVRG and evoked synchronous bilateral bursts of activity on C4 roots (Fig. 6d). Thus, rVRG V0 gluta-matergic neurons already transmit their excitation bilaterally to Ph-MNs at the time of inception of fetal breathing.

Second, we prevented the commissural navigation of the axons of V0 neurons by deleting the Robo3 gene with a Dbx1[cre] line[7]. In the absence of Robo3 receptor-mediated signaling, axons fail to navigate across the midline[20]. This conditional interference collectively targets V0 neurons of the preBötC, causing left-right de-synchronization of its activity[7], and those of the rVRG, whose role in this context had not been previously investigated. The impairment of the rVRG was attested in Dbx1[cre]; Robo3[lox/lox] E15.5 mutants by loss of contralaterally labeled rVRG neurons following a unilateral rhodamine dextran dye injection in the phrenic motor pool ($n = 3$, Supplementary Fig. 6). To investigate the functional status of this conditional mutant, we monitored the activities of facial motor nucleus (moVII) and phrenic motor output (C4). In wild-type E15.5 brainstem spinal cord prepara-tions, the activities of the moVII and C4 were bilaterally rhythmic and synchronized (Fig. 7a) and the normalized amplitudes of the left and right synchronous bouts of C4 activity were similar (1.02 $\pm$ 0.01, $n = 77$ bursts from five preparations, Fig. 7b). In Dbx1[cre]; Robo3[lox/lox] mutants the activities of the moVII were left-right de-synchronized (Fig. 7c) as if receiving only ipsilateral drives from the left-right desynchronized preBötC (Supplementary Fig. 7). Unexpectedly, at the level of the C4 outputs, bilateral synchro-nization was maintained so that each bout of moVII activity present on either side of the midline was associated with bilaterally synchronized bouts of C4 activity (Fig. 7c). However, the amplitudes of the left or right bouts of C4 activity fluctuated according to the side where the moVII, and thus probably the preBötC, were active (Fig. 7c). The normalized amplitudes of C4 discharges recorded on the active moVII side (ipsi-active) were systematically larger than those recorded from the contralateral (contra) C4, yielding an ipsi-active/contra amplitude ratio of 1.31 $\pm$ 0.09 ($n = 127$ bursts from four preparations, Fig. 7d). These unbalanced left-right C4 amplitudes are, therefore, likely to reflect an excess of descending ipsilateral projections, caused by the conditional Robo3 mutation that translates into a systematic motor overdrive on the side hosting the active preBötC. These experiments confirm the early role of V0 rVRG neurons in the mounting of the inspiratory motor command although, at this stage, commissural premotor neurons other than V0s probably also support the descending inspiratory motor drive. Interest-ingly, when we repeated these recordings in the same mutant at birth (P0) when physiological breathing needs to operate reliably, we observed a severe worsening of the unbalance of the amplitudes of the left and right C4 outputs. Indeed, the ipsi-active/contra amplitude ratio of C4 activity was dramatically increased to 4.68 $\pm$ 0.46 ($n = 76$ bursts from four preparations, Fig. 7e, f) and for some events contralateral bursts could not be detected.

To check the consequence of commissural V0 axon re-routing on the breathing behavior, we compared plethysmographic recordings of the ventilation in unrestrained wild-type and Dbx1[cre]; Robo3[lox/lox] P0 pups (Fig. 8). The mutant pups had atypical ventilation profiles (Fig. 8a, b). First, the distribution of

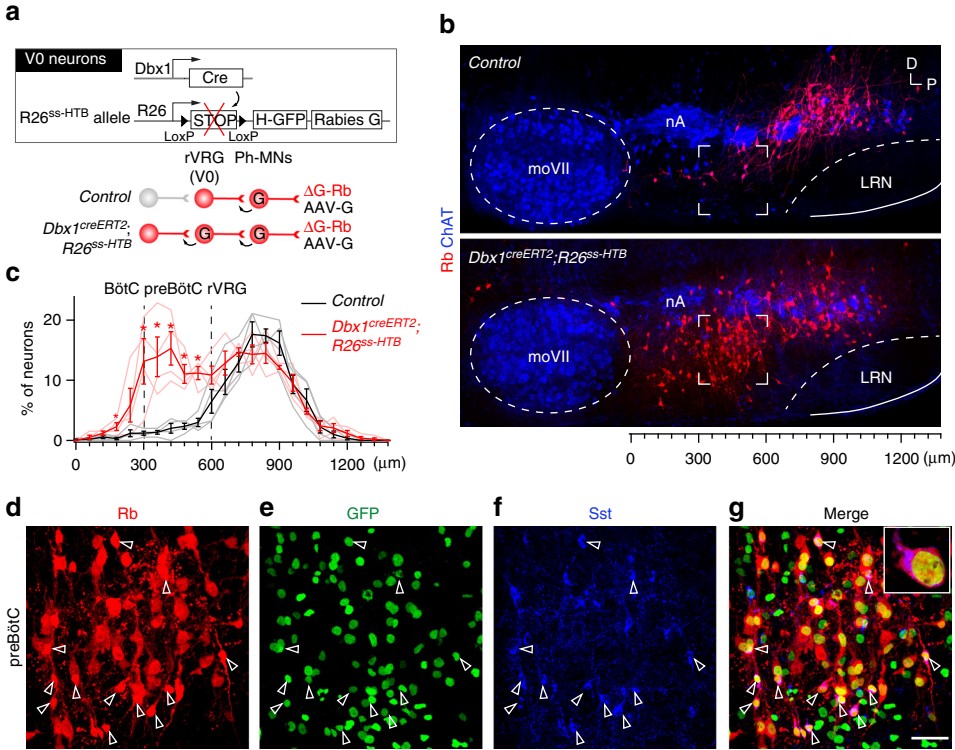

**Fig. 5** V0 homotypic connectivity from the preBötC to the rVRG. **a** Tracing scheme allowing supplementary retrograde transsynaptic spread from rVRG V0 neurons in $Dbx1^{creERT2}$; $R26^{ssHTB}$ pups. **b** Sagittal sections of the ventral respiratory column showing trace$^+$ neurons (Rb, *red*) in control (*top*) and $Dbx1^{creERT2}$; $R26^{ssHTB}$ (*bottom*) pups. Note the presence of an additional trans-synaptically labeled cell population in the PreBötC region (delineated by *square corners inset*) in $Dbx1^{creERT2}$; $R26^{ssHTB}$. **c** Summary histograms of the distributions of trace$^+$ neurons in control (*black trace*, $n = 4$) and $Dbx1^{creERT2}$; $R26^{ssHTB}$ (*red trace*, $n = 4$). **d**–**g** Zoom on the preBötC region (*inset* in **b**) of a $Dbx1^{creERT2}$; $R26^{ssHTB}$ pup showing trace$^+$ neurons (**d**) counterstained for GFP (**e**) and Sst (**f**) and overlay (**g**). *Arrowheads* report trace$^+$ neurons that coexpress GFP and Sst. *Scale bars*: **d**–**g**, 50 μm. *$p < 0.05$

breath durations ($T_{TOT}$) showed a high incidence of breath separated by abnormally short intervals (Fig. 8c). Furthermore, the distribution of tidal volumes ($V_T$) of the conditional Robo3 null mutants showed a left shift toward small amplitudes breath (Fig. 8d) so that the peak of the $V_T$ distribution in the mutant corresponded to about half that of wild-type littermates ($5.1 \pm 0.4$ μl/g, $n = 2192$ breaths events from eight pups in $Dbx1^{cre}$; $Robo3^{lox/lox}$ vs. $11.8 \pm 0.7$ μl/g, $n = 3034$ breaths events from 13 pups in wild types). This resulted in a left-shifted distribution of the ventilation measurements ($V_E = V_T/T_{TOT}$) in the mutant (Fig. 8e) owing to the large fraction of breaths characterized by both low $V_T$ and low $T_{TOT}$ values as revealed by density maps of ($V_T,T_{TOT}$) plots (Fig. 8f, g). Altogether, the shallow breathing of the mutants, that mobilized roughly half the lung capacity at about twice a higher frequency, likely results from two freely-running unilateral rhythmic motor commands targeting each a hemi-diaphragm (Supplementary Movie 1). To which extent the decoupled inspiratory drive of the MoVII might contribute to the mutant respiratory distress will need to be investigated further. Asymmetric motor drives to the left and right diaphragm are not compatible with survival, as all $Dbx1^{cre}$; $Robo3^{lox/lox}$ (13/13) newborn pups died within 24 h like Robo3 null mutants do[7, 20].

## Discussion

An early hypothesis concerning the neural bases of movements by Broadbent in 1866 stated: "That where the muscles of the corresponding parts on opposite sides of the body constantly act in concert, and act independently, either not at all, or with difficulty, the nerve-nuclei of these muscles are so connected by commissural fibres as to be pro tanto a single nucleus."[21]. This hypothesis applies well to respiratory movements controlled by "nerve nuclei" with now known origins that include the rhythm generator in the preBötC[7, 19], some of its attendant neuromodulatory control[14, 22–24] and output Ph-MNs[25]. Using transsynaptic tracing strategies we have now revealed the yet uncharted origins of a fourth essential component: phrenic premotor neurons. Among these, the principal premotor neurons that form the rVRG, share V0 identity with, and are synaptic targets of, preBötC neurons. Our data strikingly vindicate and refine the notion that hemi-diaphragms "act in concert" owing to "pro tanto a single nucleus" comprised of V0 type commissural neurons.

Our tracing scheme from the diaphragm restricts the viral spread to premotor neurons[11] and thus allowed for the first time a definitive establishment of premotor neurons locations and identities in mouse neonates. Ph-preMNs were exclusively found to reside in the pons and medulla of the brainstem and in the rostral cervical spinal cord. The bulk (about 75%) of brainstem Ph-preMNs locate to the ventral respiratory column in the BötC and, most prominently, in the rVRG. The rest of Ph-preMNs were distributed in the dorsolateral pons in the parabrachial nuclei and Kölliker-Füse nucleus (PB/KF), within the lateral tegmental area in the lateral paragigantocellular (LPGi) and gigantocellular nuclei (Gi), the intermediate reticular nucleus, in vestibular nuclei, in the midline raphe, in the dorsal medulla and in the nucleus of the solitary tract (NTS). Altogether, these locations are in agreement with previous anatomical and electrophysiological delineations of Ph-preMNs made in the adult mouse[9], rat[8] and cat[10], suggesting that inspiratory descending circuits are conserved in mammals and definitively set 1 week after birth. The relatively few Ph-preMNs outside the ventral respiratory column may reflect modulatory roles: integration of

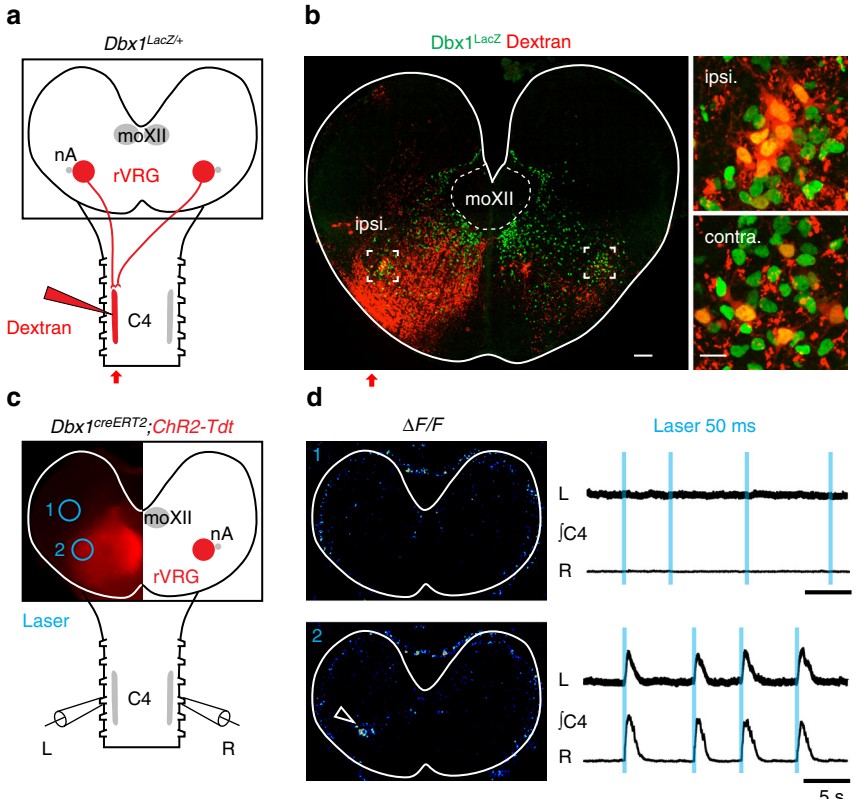

**Fig. 6** Neurons of the rVRG transmit bilaterally their excitation to Ph-MNs at E15.5. **a** Retrograde tracing scheme in a *Dbx1^LacZ/+* brainstem spinal cord at E15.5 using Rhodamine dextran dye unilaterally injected in at C4 level. **b** *Left*, transverse slice showing the tracer pattern (*red*) and LacZ counterstain (*green*) in the ipsi- and contralateral rVRG (*insets*) to the tracer injection side. *Right*, zoom on the *insets* showing double labeled (*yellow*) rVRG cells. **c** Schematic showing the *Dbx1^creERT2; ChR2-Tdt* preparation and unilateral illumination targets (*blue empty circles*) achieved by computer generated holography outside (1) and on (2) the rVRG while recording activities of *left* and *right* C4 motor roots. **d** Top photostimulation away from the rVRG (*blue circle* 1 in **c**) fails both to trigger detectable Δ*F/F* fluorescence changes and evoke C4 activity responses (*right* set of traces). *Bottom*, the light spot positioned on the rVRG (*blue circle* 2 in **c**) evokes Δ*F/F* fluorescence changes restricted to the targeted rVRG (*arrowhead*) and synchronous *left* and *right* C4 bursts (*right* set of traces). Note that the photostimulation of the rVRG on one side fails to activate the contralateral rVRG. *Scale bars*: **b** *left*, 100 μm; **b** *right*, 20 μm

viscero- and chemo-sensory afferents in the NTS[26], inputs from higher brain centers coordinating various rhythmic motor activities in the PB/KF[27, 28] or necessary persistence of breathing during REM sleep atonia through the LPGi[29]. All these premotor locations featured ipsi-laterally biased projections to Ph-MNs that could introduce an asymmetry of the phrenic premotor drive should they be activated unilaterally. To our knowledge asymmetric volitional breathing has been attested in human patients with vascular hemiplegia[30], suggesting that corticospinal descending inputs thought to indirectly project on Ph-MNs[31] may target first such premotor neuronal populations.

The part of the ventral respiratory column that extends caudally from the BötC to the rVRG plays the major role in the motorization of the inspiratory pump. It hosts the vast majority of traced Ph-preMNs that are evenly distributed across the midline following unilateral viral injections (Supplementary Table 1). Excitatory and inhibitory Ph-preMNs originate from two distinct sources of progenitors and differentially populate the BötC and the rVRG. Inhibitory vGAT⁺ Ph-preMNs have a dB1/dB4 origin, form all of the BötC and contribute a third of rVRG neurons, while excitatory vGlut2⁺ Ph-preMNs by large of p0 origin reside in the rVRG. The few Ph-preMNs located in the area of the preBötC are indistinguishable from rVRG neurons by molecular criteria, suggesting a minor, if any, contribution of the preBötC in the direct transmission of descending drive to Ph-MNs[8]. Our work identified a fraction of rVRG neurons as

inhibitory neurons that together with rVRG excitatory ones may provide concurrent glutamatergic and GABAergic drives to, (i) enable a specific gain control of the Ph-MNs output during inspiration in a BötC-independent manner[32], (ii) organize the recruitment of Ph-MNs or smooth out diaphragm force production[32, 33]. Finally, the inhibitory Ph-preMNs of the BötC are known to provide the synaptic inhibition of Ph-MNs during expiration[34].

One major finding is that the predominant premotor group, the glutamatergic rVRG neurons have for the most part V0 identity, like the preBötC. The chief importance, collectively, of these two classes of V0 neurons in breathing coordination is revealed by disrupting their commissures, which caused left-right desynchronized breathing at birth incompatible with survival. The same genetic interference resulted at E15.5 in milder unbalance of left and right C4 drives, showing that the phrenic premotor organization is initially less dependent on V0 cells. Unlike motor behaviors that mature through postnatal incorporation of novel premotor modules[35], the inspiratory circuits thus seem to undergo a weeding out of the connectivity outside the preBötC-rVRG module, present at the inception of fetal breathing (this study and ref.³).

A novel trait, not revealed by traditional tracing methods, is that Ph-preMNs display a special axonal morphology: a bilaterally branched axon synapsing onto phrenic motor pools on both sides of the midline. For technical reasons (see material and methods)

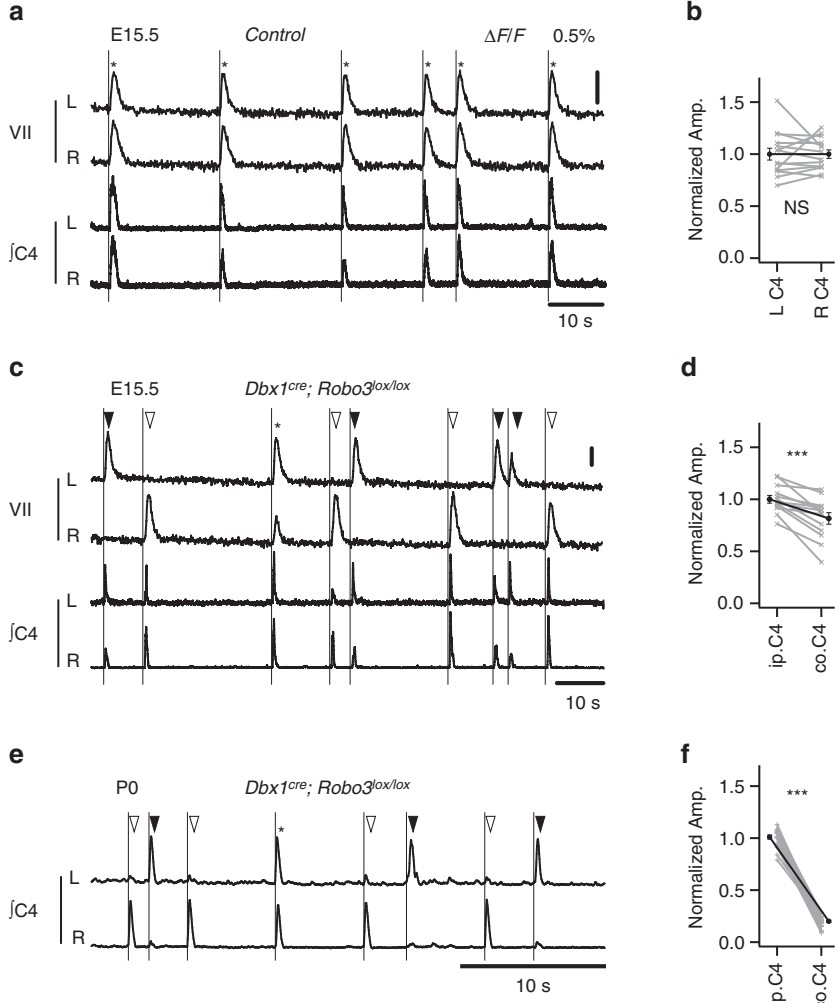

**Fig. 7** V0 neurons ensure bilaterally synchronized and balanced C4 outputs in vitro. **a** Spontaneous fluorescence ($\Delta F/F$) changes of the *left* (L) and *right* (R) facial motor nucleus (VII, *top traces*) and electrophysiological activities of the *left* and *right* C4 motor root (*bottom traces*) showing phased (*) rhythmic bouts of activity in wild-type preparations. **b** Histogram of the normalized amplitudes of synchronous *left* (L C4) and *right* (R C4) phrenic motor bursts showing their balanced amplitudes. **c** In *Dbx1cre; Robo3lox/lox* preparations, the *left* (*black arrowheads*) and *right* (*white arrowheads*) bouts of VII activity were found de-synchronized but still phased (*vertical lines*) with bilaterally synchronized bouts of activity of the C4 motor roots. Note the higher amplitude of the burst on the C4 root ipsilateral (ip.C4) to the active VII when compared to that recorded from the contralateral side (Co.C4). **d** Histogram of the normalized amplitudes of synchronous ip.C4 and Co. C4 showing the reduced amplitude of the latter. **e** *Dbx1cre; Robo3lox/lox* mutant preparations, at P0, show an aggravated unbalance of the amplitudes of the synchronous bouts of activity of the *left* and *right* C4. **f** Histogram of the normalized amplitudes of yet synchronous ip.C4 and Co. C4 bursts of activity showing the much greater amplitude of ip.C4 relative to Co. C4 bursts suggesting an almost complete loss of commissural connectivity. Phased activity events (*) were still detected occasionally in *Dbx1cre; Robo3lox/lox* preparations at both E15.5 (17/144 bursts from four preps) and P0 (6/79 bursts from four preps) either caused by fortuitous phasing of the independent *left* and *right* motor drives or by a yet unidentified source of synchronous bilateral drives to the rhythm generator. ***$p < 0.001$

these neurons are probably underestimated in our study. Yet, this axonal profile was detected in all of the main phrenic premotor areas: the PB/KF, the NTS, the BötC and in the rVRG. Such a feature has been previously described for components of the premotor circuits that control bilateral whisking[35] and jaw closing movements[36]. Our data indicate that this morphological trait is not related to neuronal origin or subtype identity. Indeed, such neurons were present in premotor stations that originated in dorsal dA, dB domains, in the ventral p0 domain of the neural tube as well as in, necessarily, a fraction only of dA3 subtype neurons of the NTS found with predominant ipsilateral projections. The yet unknown signal that produces axon collateral branching, appears a crucial morphogenetic instruction in several brainstem premotor circuits to ensure the simplest design for bilateral coordination: a single premotor neuron that synapses on equivalent ipsilateral and contralateral motor neurons.

Our tracing experiments also revealed that, collectively, Ph-preMNs send collateral projections to several brainstem visceral and somatic motor nuclei that innervate oro-facial and upper airway patency regulating muscles. However, these projections did not arise from rVRG Ph-preMNs. In fact, axonal terminals of rVRG neurons were only manifest in the LRN, a pre-cerebellar nucleus that, through mossy fibers, relays information from several spinal systems controlling posture, reaching, grasping, and locomotion[37]. This is consistent with the finding that spinal premotor neurons underpinning voluntary forelimb movements and rVRG premotor neurons directly connect to LRN neurons[18]. Altogether these findings suggest that the rVRG is the main source of inspiratory corollary discharge[38] needed to coordinate breathing with other motor, e.g., postural, locomotor, feeding, expulsive behaviors that rely on partially shared muscles[39]. Importantly, V0 rVRG axonal terminals were missing in the

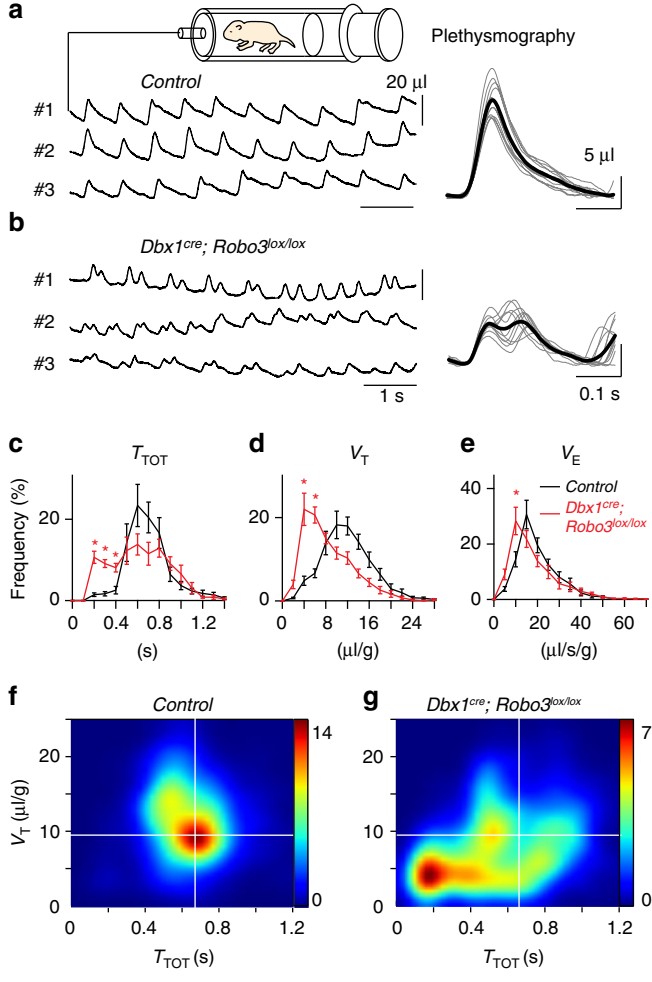

**Fig. 8** Left-right decoupled breathing in $Dbx1^{cre}$; $Robo3^{lox/lox}$ neonates. **a** *Left*, example plethysmographic recordings from three wild-type P0 neonatal pups (#1, #2, #3) during quiet breathing. *Right*, ten superimposed representative breath (*thin lines*) and their average (*thick line*). **b** *Left*, comparable recordings for three $Dbx1^{cre}$; $Robo3^{lox/lox}$ mutant pups. Note the reduced amplitude of breaths and their more frequent occurrence. *Right*, superimposed representative mutant breaths often appearing as double breath events. **c**–**e** Summary histograms of the distributions of breath duration ($T_{TOT}$, **c**), tidal volumes ($V_T$, **d**) and ventilation ($V_E$, **e**) in wild types (*black trace*), $Dbx1^{cre}$; $Robo3^{lox/lox}$ (*red trace*). Note the *left* shifted distributions for $T_{TOT}$, $V_T$, and $V_E$ in mutants. *Stars* indicate significant differences with wild types. **f**, **g** Density maps of $V_T$–$T_{TOT}$ relationships for the breathing of wild types (**f**) and $Dbx1^{cre}$; $Robo3^{lox/lox}$ (**g**). Note in the mutants the high density of aberrant breath featuring both small $V_T$ and small $T_{TOT}$ values (in the *bottom left* quadrant) absent in wild-type pups

rVRG itself, a situation strikingly contrasting with preBötC V0 neurons whose intrinsic ipsi- and contra-lateral connectivities ensure rhythm generation and left-right synchronicity.

In sum, our results reveal a core executive control circuit for inspiration comprised of a V0 bilaterally synchronized rhythm generator feeding a V0 bilaterally projecting V0 rVRG premotor station. This redundant architecture alleviates the possibility that asymmetric inputs to this core may translate in unbalanced motor drives to the left and right hemi-diaphragms.

A number of studies hint at the possibility that transcription factors may induce expressing neurons to establish synaptic connections with one another during development, to form

neural networks. This may concern $Tlx3$[40], $Brn3a$[41], $Drg11$[42] in the somatic sensory pathways, $Atoh1$ in proprioceptive pathways[43], $Lhx6$ in an amygdalo-hypothalamic pathway[44] and $Phox2b$ in visceral circuits[45]. V0 neurons in the preBötC and the rVRG recapitulate all of the necessary properties for generation and transmission of the rhythmic inspiratory command in a secured bilaterally synchronized manner to Ph-MNs, for effective muscle contraction powering breathing inflow. Thus, synapses made by neurons sharing a common V0 identity, are sufficient for building an inspiratory motor circuit during development and, in a more speculative way, for the advent of aspiration breathing in tetrapods[46].

All extant amniotes draw air in by expanding the thorax (aspiration breathing) as opposed to amphibians, who pump air into the lung by building up pressure in the bucco-pharyngeal cavity[47]. The amniotes that first evolved a diaphragm required, for its motorization, that respiratory activity be shifted from cranial branchiomotor nerves to spinal somatic ones (e.g., the phrenic nerve), and importantly, that a neural circuit controlling its contractions arose. The present data support an evolutionary scenario whereby these demands may have been met simultaneously. In the spinal cord, V0 neurons were first described as commissural inhibitory premotor neurons synapsing onto somatic motor neurons that innervate hindlimb muscles[48]. Assuming that this is the default state of V0s produced throughout the neuroaxis, those that are produced in the rVRG (this study) and in the preBötC[7] differ from their serial spinal homologues by a glutamatergic phenotype (but see ref. [49]). And among those, the latter are rhythmogenic and not premotor, while, as shown here, the former are premotor and not rhythmogenic. We hypothesize that during evolution two neighboring groups of hindbrain commissural premotor V0 neurons simultaneously acquired a glutamatergic phenotype while the rostral group only, the preBötC (possibly through a rhombomere specific-mechanism), acquired the capacity to connect, rather than to motor neurons, to other V0 neurons, i.e., to form homotypic V0-V0 synapses. This would suffice to jointly establish, on the one hand, the preBötC as a rhythm generator whose function relies on the synaptic interconnections of its constitutive V0 neurons according to the group pacemaker hypothesis (ref. [50] and see ref. [1] for a review), and on the other hand, the V0 premotor rVRG as its target. Thus, two subtle V0 fate changes would enable inspiratory rhythm generation and direct its output to Ph-preMNs, thus establishing a redundantly commissural V0 core circuit to secure the bilaterally coordinated contractions of inspiratory pump muscles required for aspiration breathing.

## Methods

**Mouse genetics.** The following previously described mouse lines were used in this study: $Dbx1^{cre}$ (ref. [51]), $Dbx1^{LacZ}$ (ref. [52]), $Dbx1^{creERT2}$ (ref. [53]), $En1^{cre}$ (ref. [54]), $Lbx1^{cre}$ (ref. [13]), $Sim1^{cre}$ (ref. [55]), $VGlut2^{cre}$ (ref. [56]), $vGAT^{cre}$ (ref. [57]), $ChAT^{cre}$ (ref. [58]), $Robo3^{lox/lox}$ (ref. [59]), RCL-ChR2(H134R)/EYFP abbreviated Ai32 and RCL-hChR2(H134R)/tdT-D abbreviated Ai27[60], Tau-lox-stop-lox-Syn-GFP-IRES-nlsLacZpA[61], $R26^{ssHTB}$ (ref. [62]). For recombination induction, Dbx1creERT2 mice were orally gavaged with Tamoxifen (20 mg/ml in corn oil, Sigma) at E10.5 at a dose of 0.1 mg/g of body weight. Live embryos were recovered at E18–E19 through cesarean section, fostered, and raised for further analysis. All animal studies were done in accordance with the guidelines issued by the European Community and have been approved by the research ethics committees in charge (Comité d'éthique pour l'expérimentation animale) and the French Ministry of Research. Sample sizes were chosen on the basis of previous literature. No method of randomization was used to determine how animals were allocated to experimental groups and the investigators were not blinded when analyzing data.

**Retrograde tracing experiments.** Production of Rb-mCherry, Rb-GFP, and AAV-G for virus experiments with monosynaptic restriction to label premotor neurons were carried out as previously described[11, 63]. P1 mouse pups of a given Cre-line, were anesthetized by hypothermia, prior to receiving viral injections targeted uni- or bi-laterally to the costal part of diaphragm muscle. Two microliters

of a viral solution containing equal volumes of G-deficient rabies virus (titers of ~ 1e + 8) and AAV-G of serotype 6 (titers ~ 3e + 12) were injected in individual muscles. Eight days post injection, P9 pups were deeply anesthetized, transcardially perfused with 4% paraformaldehyde (PFA) in phosphate-buffered saline (PBS), post-fixed overnight in 4% PFA, and cryoprotected in 30% sucrose in PBS and were stored at −80 °C for later immunohistochemistry (IHC). To ascertain that the tracing scheme targeted centrally motor neurons to initiate the viral spread, we compared the distribution of premotor neurons obtained with the above tracing scheme with that resulting from comparable injections using EnvA-pseudotyped ΔG-Rb (titers of ~ 1e + 9) in $ChAT^{cre}$; $R26^{ssHTB}$ animals. The latter tracing scheme was found less effective than the original one. The total number of traced Ph-preMNs (counted in $n = 2$ experiments) was found reduced by about 35% probably owing to suboptimal viral infection of seeding Ph-MNs in relation to limited cre-dependent expression of TVA from the $R26^{ssHTB}$ allele.

To allow a supplementary retrograde transsynaptic jump of ΔG-Rb from rVRG neurons, we performed unilateral diaphragm injections of viral solution (G-deficient rabies virus and AAV-G) in the $Dbx1^{creERT2}$; $R26^{ssHTB}$ genetic background.

Retrograde tracing in E15.5 preparations was performed using pressure injection of Rhodamine dextran MW 3000 (Molecular probes) in the C4 ventral spinal cord of isolated brainstem spinal cord preparations in vitro. After injection, preparations were maintained in the perfusing chamber with the oxygenated (95% $O_2$, 5% $CO_2$) artificial cerebrospinal fluid at RT for 12 h prior to fixation in 4% PFA. Transverse and sagittal 20–60 μm thick sections of the entire brain were performed using a cryostat (Leica CM3050, Germany).

**Histology, imaging, and cell counting.** The methods for IHC, combined bright field in situ hybridization (ISH) have been described previously[7, 23]. *Gad1* and *GlyT2* riboprobes were synthesized using a DIG RNA labeling kit (Roche, Germany) as specified by the manufacturer. And all ISH signals were revealed by NBT and BCIP (Roche). Antibodies used for IHC were as follows: Chicken anti-GFP (1/2000, Aves Labs GFP-1020), rabbit anti-GFP (1/2000, Invitrogen A11122), chicken anti-beta-galactosidase (1/2000, Chemicon AB3403), goat anti-ChAT (1/100, Millipore AB144P), rabbit anti-RFP (1/1000 Rockland 600-401-379), rat anti-RFP (1/1000, ChromoTek 5F8), rabbit anti-Pax2 (1/300, Covance PRB-276P), rabbit anti-NK1R (1/5000, Sigma S8305), rabbit anti-Sst (1/500, Peninsula T4103), rabbit anti-Tlx3 (1/5000, kind gift from C. Birchmeier), rabbit anti-Phox2b (kind gift of JF Brunet), mouse anti-Islet1,2 (1/50, DSHB 39.4D5c and 40.2D6c). The primary antibodies were revealed by alexa 488- (Invitrogen, Carlsbad, CA), DyLight 405-, Cy3-, alexa 594-, DyLight 649-, or Cy5-labeled (Jackson Immunoresearch, Suffolk, UK) secondary antibodies of the appropriate specificity. Slides were visualized using a Leica SP8 confocal microscope.

The virally infected premotor neurons were carefully surveyed in the entire brain rostral to the spinal cord. Neurons were counted on both sides in all 1:1 serial sections. Results are expressed as mean ± SEM. The number of traced Ph-preMNs was variable (range: 401–792 neurons; 577 ± 62 neurons, $n = 6$) from one injection to the other but the relative abundance of neurons in the diverse premotor areas was consistent between experiments. Experiments that resulted in trans-synaptic labeling of less than 150 rVRG neurons ($n = 3$) failed to label neurons in any other premotor location and were discarded. Neuronal counts in premotor groups are expressed as percentage of the total number of trace$^+$ premotor neurons within the sample. In bilateral diaphragm injection experiments with red and green viruses, the number of doubly labeled premotor neurons is influenced by the probability that primo-infection occurred in pairs of left and right Ph-MNs commonly targeted by a presynaptic partner and by possible competition of the two viral vectors for expression in co-infected cells. This probably introduces an underestimation bias that precluded quantification of Ph-preMNs bearing bilaterally descending branched axons. To quantify secondary viral spread from rVRG neurons in $Dbx1^{creERT2}$; $R26^{ssHTB}$ conditional mice while taking into account the inherent variable efficiencies of infections, neuronal counts were expressed as percentage of total number of trace$^+$ rVRG neurons within the sample.

We use the following nomenclature for brainstem structures: rVRG (rostral ventral respiratory group), PB/KF (Parabrachial nuclei and Kölliker-Füse nucleus; given the lack of defined boundaries we pooled lateral and medial subdivision of the parabrachial nucleus), preBötC (preBötzinger complex), BötC (Bötzinger Complex), NTS (nucleus of the solitary tract), LPGi, Gi (Gigantocellular reticular nucleus) Ve (includes vestibular nuclei, medial, lateral and spinal parts), IRt (Intermediate reticular formation), Raphe (includes raphe magnus and obscurus).

**Location of the BötC, preBötC and rVRG in neonatal mice.** In neonatal mice, at P9, the three subdivisions of the ventral respiratory column: the BötC, preBötC and rVRG, were defined using previously determined anatomical criteria[64, 65]. From rostral to caudal, the BötC was located caudal to the end of facial motor nucleus (moVII) and ventral to the compact formation of nucleus ambiguus (nAc) (Supplementary Fig. 4a). The preBötC caudal to the BötC and ventral to the semi-compact formation of nucleus ambiguus (nAsc), started 300–350 μm caudal to posterior end of moVII (Supplementary Fig. 4b). The rVRG, caudal to preBötC, extended from 550–600 μm caudal to the posterior end of moVII to the rostral pyramidal decussation, ventrally to the loose formation of nucleus ambiguus

(nAls), intermingled with the external formation of nucleus ambiguus (nAex) and dorsal to the LRN (Supplementary Fig. 4c).

**Digital three-dimensional brainstem reconstructions.** The 3D brainstem reconstruction of the virus-labeled premotor neurons was processed as previously described[66]. Briefly, all sections were acquired with a confocal microscope (Olympus) using a 20× objective. In order to cover the full area of the brainstem with labeled premotor neurons, a mosaic 7 × 5 tile was acquired for each brain section, and stitched by Fiji software. All stitched images were aligned manually using Amira software to construct the 3D model, in which rabies labeled (trace$^+$) premotor neurons were manually assigned (Imaris Spot Detection), and color-coded according to their location based on Paxinos' mouse brain atlas[67].

**Calcium imaging, electrophysiology, and photostimulation.** The methods used for preparing mouse brainstem–spinal cord preparations from embryonic day 15.5 embryos and P0 mouse and maintaining them in oxygenated artificial cerebrospinal fluid (a-CSF) have been described[7, 23, 24]. Briefly, brainstem-spinal cord preparations were dissected in 4 °C a-CSF of the following composition (in mM): 128 NaCl, 8 KCl, 1.5 $CaCl_2$, 1 $MgSO_4$, 24 $NaHCO_3$, 0.5 $Na_2HPO_4$, 30 glucose, pH 7.4. For calcium imaging of MoVII neurons, brainstem-spinal cord preparations were incubated at room temperature for 40–45 min in oxygenated a-CSF containing the cell-permeable calcium indicator dye Calcium Green-1 AM (10 μM; Life Technologies, Paisley, UK) and were transferred to a recording chamber (30 °C). Optical recordings started after rinsing out for 30 min the excess of dye using a conventional epifluorescence configuration with a FITC filter cube. Fluorescence images were captured from the ventral surface of brainstem-spinal cord preparations exposing the MoVII region, with a cooled Neo sCMOS camera (Andor Technology Ltd., Belfast, UK) using 4× objectives, 4 × 4 binning, an exposure time of 100 ms and for periods of 180 s using Micro-Manager software (https://www.micro-manager.org/wiki/).

Phrenic nerve activity was recorded on E15.5 and P0 brainstem-spinal cord preparations using suction electrodes positioned on the fourth cervical root (C4) as described[7, 23, 24]. The raw signals were amplified (High-gain AC, 7P511, Grass Technologies, Warwick, RI), filtered (bandwidth 0.1–3 kHz) and integrated (time constant 50 ms, Integrator 7DAEF, Grass Technologies, Warwick, RI) before digital sampling at 6 kHz and analysis using pClamp9 (Molecular Devices). Values are given as mean ± SEM. Statistical significance was tested using a paired difference Student's $t$-test to compare the measurements obtained in two different conditions.

In optogenetic experiments, photostimulation was performed using computer generated holography[68]. Briefly, a 473 nm DPSS laser (CNI, Changchun, China) was used to excite ChR2-expressing neurons. The output beam was expanded to match the input window of a spatial light modulator LCOS-SLM (X10468-01, Hamamatsu), operating in reflection mode. A custom-designed software calculated, given an intensity distribution at the focal plane of the microscope objective, the phase hologram and addressed it to the LCOS-SLM[68]. The SLM plane was imaged at the back aperture of the microscope objective through a telescope. Individual light pulses of 50 ms duration (laser power density 1–5 mW/mm$^2$) were delivered manually at 7–10 s intervals onto a 100 μm circular region covering the rVRG exposed transversally in E15.5 preparations (Fig. 6c) while recording the electrophysiological activities from the left and right C4 motor roots.

**Commissural interference.** For *Robo3* conditional deletion experiments using $Dbx1^{creERT2}$ mice, a single dose of tamoxifen at E10.5 failed to alter bilateral synchrony of activities in the preBötC in transverse slices (data not shown). Extending the treatment to optimize recombination by daily injections over 3 days starting on E9.5 precipitated abortions and yielded altogether only one E15.5 embryo with attested C4 bilateral amplitude unbalance. As tamoxifen treatment, tested in wild-type pregnant females, often resulted in pups showing signs of respiratory distress, we resorted to the use of the non-inducible $Dbx1^{cre}$ line for both prenatal and P0 functional investigations.

To compare the amplitude of activities recorded from the left and right C4 motor roots in wild-type preparations, we first normalized the amplitude of individual bursts on each side to the mean amplitude of all burst recorded on the corresponding side during the recording period. We then calculated the ratios of the normalized amplitudes of synchronous left and right bursts. To compare the amplitude of C4 activities on either side of the midline in conditional *Robo3* null mutants at E15.5, we first normalized the amplitudes of C4 burst on each side to the mean amplitude of all the C4 bursts ipsilaterally co-active with the MoVII (ipsi-active) on the corresponding side. Then the normalized amplitudes of ipsi-active bursts were divided by that of C4 bursts simultaneously recorded on the opposite side (contra) to calculate ipsi-active/contra amplitude ratios. At P0, due to the large unbalance of the amplitudes of left and right C4 bursts, largest peaks were assumed to arise in the ipsi-active side and their amplitude considered as the numerators for calculating ratios.

**Plethysmography.** Breathing variables of un-anesthetized, unrestrained P0 pups were measure by whole-body barometric plethysmography as previously described[23]. After a 7 min adaptation period in the plethysmograph chamber, the breathing parameters (breath duration ($T_{TOT}$), tidal volume ($V_T$), and ventilation

($V_E$) calculated as $V_T/T_{TOT}$) were continuously monitored in apnea free periods and scored using a custom software package (Elphy by G Saddoc, https://www.unic.cnrs-gif.fr/software.html). Values are given as mean ± SEM. After using F-test to compare the equality of variances, statistical significance was tested using an unpaired difference Student's $t$-test to compare data sets obtained from control and mutant pups. Density maps of breath amplitude and duration ($V_T$, $T_{TOT}$) relationships were obtained using a Python custom made plugin.

**Data availability.** All relevant data that support the findings of this study are available from the corresponding author upon reasonable request.

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

## Acknowledgements

This work was supported by grants from the Agence Nationale de la Recherche (ANR-10-BLAN1410-02 and ANR-32-BSV5-0011-02 to G.F.), Fondation pour la Recherche Médicale (DEQ20120323709 to G.F.), P.C. and S.A. were supported by the Kanton Basel-Stadt, the Novartis Research Foundation, an ERC Advanced Grant and a Swiss National Science Foundation grant, J.W. was supported by fellowships from Fondation pour la Recherche Médicale and from Ecole des Neurosciences de Paris. M.G. was supported by NIH grants NS090919 and NS080586.

## Author contributions

J.W. was involved in design of experiments, carried out experiments, acquired and analyzed data. P.C. performed 3D-reconstructions and produced viral vectors. J.B. and M.G. contributed some of the early data. S.A., G.F. conceived experiments. G.F. supervised the project and wrote the manuscript. All of the authors discussed the results and implications and commented on the manuscript at all stages.

## Additional information

**Competing interests:** The authors declare no competing financial interests.

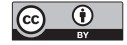

