## [Peer Review File · Nature Communications]

Reviewers' comments:

Reviewer #1 (Remarks to the Author):

All behavior requires movement, with the obvious consequence that the critical structures in the nervous system producing the behavior must be connected to the relevant motoneurons. For breathing, the primary inspiratory muscle is the diaphragm, innervated by phrenic motoneurons. Since the inspiratory rhythm generator is the preBotzinger Complex (preBötZ) there must be a rather direct pathway between the two that is mostly represented by premotor neurons in the rostral ventral respiratory group (rVRG). Here the authors use a contemporary, highly sophisticated viral-based track tracing technique in mice to identify the location and phenotype of inspiratory premotor neurons and to delineate some of their properties and projections. The study is done with admirable care and precision, and the data is impeccable and valuable. The authors considerably advance previous studies by defining the transcription lineage of premotor neurons, conclusively establishing their transmitter phenotypes and the laterality of their projections, and show the importance of the V0 lineage in the connections between the preBötZ and rVRG. One important conclusion is that the majority of vGlut2+ Ph-preMNs resides in the rVRG where they appear to have an exclusive V0 identity. This and their other interpretations are fair and reasonable and represent a significant advance in understanding the motor control of inspiratory movements. Most of my comments related to the INTRODUCTION and DISCUSSION, which are mostly minor.

SUMMARY

"These motor drives emerge from interactions between critical sets of brainstem neurons whose identities and synaptic ordered organization remain unresolved." I don't understand "unresolved". Certainly, there are numerous published papers. What is "unresolved"?

"...the principal inspiratory premotor neurons share V0 identity with, and are connected by, neurons of the preBötzing complex that pace inspiration." While the preBötZ paces inspiration, the preBötZ to rVRG projections may not be from the neurons that actually "pace" inspiration.

"Deleting the commissural projections of V0s results in left-right desynchronized inspiratory motor commands in reduced brain preparations and breathing at birth." Species?

INTRODUCTION

"In mammals, breathing is a motor behavior generated by a central pattern generator (CPG) located in the brainstem". CPGs usually include motor neurons, so the respiratory CPG is in the brainstem AND spinal cord.

"...allowing for breathing practice period prior to the challenge of encountering air at birth." Typo.

"The CPG respects two intangible constraints, namely the synchronicity and the balanced amplitude..." balanced amplitude ??? Is this true in all cases for all inspiratory muscles?

"The identity of neurons in charge of ensuring fail-safe bilaterally synchronized and amplitude balanced inspiratory motor drive is investigated here." What about this work addresses "fail-safe"?

P4 "However, nothing is known of the identities of premotor neurons" Nothing? Isn't there considerable literature delineating the location, afferent and efferent projections, transmitter phenotype, etc. of these neurons. In the DISCUSSION, the authors acknowledge this: "Altogether,

these locations are in agreement with previous anatomical and electrophysiological delineations of Ph-preMNs made in the adult mouse 23, rat 24 and cat 25," but without a similar statement earlier, the naïve reader may think "nothing" is known.

"and of the specific synapses downstream the rhythm generator that secure temporal and amplitude patterning of the inspiratory motor drive", More precision is needed here. Certainly, we know that the preBötz projects to rVRG premotor neurons that in turn project to the diaphragm (see previous comment). The value added in this paper relates to further refinement of the neuronal phenotypes. Also typo "downstream OF..."

"Here we addressed this question using viral-based circuit-mapping approaches 8 from the diaphragm muscle in early postnatal mice. We find that phrenic premotor neurons (Ph-preMNs) are distributed at several sites of the brainstem and include neurons with bifurcating axons that connect to phrenic motor neurons (Ph-MNs) on both sides of the midline." Dobbins et al (REF 24) used "viral-based circuit-mapping" approaches in another rodent species (rat) and basically had the same results as described here, as the authors ultimately acknowledged in the DISCUSSION. The authors need to emphasize the novelty of their approach, and their new/novel/more refined findings.

RESULTS

P4 "To selectively label neurons that synapse onto Ph-MNs, we used transsynaptic rabies technology with monosynaptic restriction." The actual technique used provides significantly more precision than prior use of "transsynaptic rabies technology WITHOUT monosynaptic restriction" (REF 24). It would be useful for the authors to acknowledge this and point out what is novel here.

P5, bottom: How do these results compare to REF 24 in rats? This is in the DISCUSSION, but a comment here might be useful.

P8 "Taken together, our data establish that the majority of vGlut2+ Ph-preMNs resides in the rVRG where they have an exclusive V0 identity." Need to rephrase. This may be true since a similar proportion of trace+ neurons are VGlut2+ or Dbx1+ with a similar expression of Pax2, but no data showing that all VGlut2+ Ph-preMNs in the rVRG are Dbx1+. Rather, it would be fair to say that trace+ neurons of V0 origin appear exclusively in the rVRG (and preBötz).

P9 "We first verified the massive presence of rVRG neuron synaptic terminals (7.7 ± 0.52 boutons/soma, $n=62$ cells) on motor neurons of the Ph-MN pool (Fig. 4a-c)." This was previously shown (without quantification or such sophistication) by Feldman et al. J. Neurosci, 1985.

P9 "Together these data indicate that rVRG neurons are not intrinsically connected and that they send collateral projections to the LRN but not to major respiratory-related cranial motor neurons." Need to rephrase. Fig. 4d-f shows that V0 trace+ rVRG neurons do not synapse onto the somas of other trace+ rVRG neurons, not that the rVRG lacks interconnectivity or only projects to the LRN. Also in Fig. 4g there appears be yellow V0 trace+ puncta in the projection field dorsal to the LRN. Could there be synapses onto trace- rVRG neurons?

P12 "We conclude that two sets of Dbx1-derived excitatory neurons redundantly ensure (i) generation of a bilaterally synchronized inspiratory rhythm and (ii) the ... synchronized premotor drives required for optimal efficiency of the respiratory pump at birth." Substantial conclusions are best in the DISCUSSION, not in the RESULTS.

DISCUSSION

P 13 "Using trans-synaptic tracing strategies we have now revealed the yet uncharted identities of a fourth essential component: phrenic premotor neurons." As noted above, the authors reveal novel aspects (is this what "uncharted identities" mean?) of this projection, but give the impression that almost "nothing" was known prior to this work (as so stated earlier by the authors, commented above).

P13-14 "excitatory vGlut2+ Ph-preMNs have a PO origin and almost exclusively reside in the rVRG." Typo "VO", and rephrase since not shown that all vGlut2+ Ph-preMNs are Dbx1+.

P15 "One major finding is that the predominant premotor group, the glutamatergic rVRG neurons have a VO identity.." Same as above

P 16 "A novel trait, not revealed by traditional tracing methods, is that Ph-preMNs display a special axonal morphology: a bilaterally branched axon projecting onto phrenic motor pools on both sides of the midline." This was previously shown in other species, so the novelty is the identification in mice not the finding itself.

SUPPLEMENTAL FIGURES

SUPP Fig. 1e Trace+ neurons appear to be in the supratrigeminal nucleus rather than the PB/KF- consider using another representative image.

Reviewer #2 (Remarks to the Author):

In the study by Wu and colleagues the authors investigated the developmental origin and anatomical organization of neural populations and circuits that control breathing. Using rabies virus-mediated transsynaptic labeling on genetically defined populations of brainstem neurons, the authors characterize the neuronal subtypes targeting phrenic motor neurons. The authors show that the predominant population of premotor neurons targeting phrenic motor neurons are located in the rostral ventral respiratory group. Similar to the rhythmically active neurons in preBotzinger complex, rVRG neurons are derived from interneurons that express the transcription factor Dbx1. They show that these rVRG neurons are glutamatergic, and VRG neurons target both ipsilateral and contralateral MN populations. Using Robo3 conditional mutants to disrupt the commissural projections of VO neurons, the authors characterize the function of VRG neurons in coordinating bilateral muscle contraction.

Previous studies by the authors and others have characterized the developmental origin of the central pattern generating networks in the brainstem, as well as the effector motor neuron subtypes that drive respiratory muscle. How brainstem networks engage respiratory motor neurons is less clear, and this study provides an important link between these two systems. The authors make the interesting observation that both preBotz and VRG neurons derive from populations expressing the Dbx1 transcription factor.

Overall this is an excellent paper, and provides a major contribution to our understanding of the basic architecture of the neural circuits that control breathing. The study beautifully characterizes anatomical organization of prePhrenic MNs. The results are easy to follow and the data are well quantified. The paper generally well written, with the exception of the last two data figures, which need minor revision (see below).

Major comments.

1. In addition to phrenic motor neurons, respiratory function requires coordination of intercostal and abdominal muscle. Presumably in amniotes which lack a diaphragm (such as in birds and reptiles) axial muscles provide the major inspiratory drive. Can the authors comment on this? For example, do rVRG neurons also target other MN subtypes in addition to the phrenic as shown in Figure 4a-c?

2. Also the evolutionary arguments in the discussion imply the breathing circuits were initially for "Motorizing the diaphragm". But the diaphragm has only been described in mammals, so it seems that there were at least two independent events (first a PreBotz-VRG-axial respiratory circuit; then later a VRG-phrenic). I am unsure how well this has been studied in chick or other model system, but it seems the evolutionary history is more complex than stated.

3. I would also suggest the authors consider restating the argument in the last paragraphs of the discussion again for clarity. I think the hypothesis about two distinct steps in VO diversification is an interesting hypothesis, but the way it is written is hard to follow. If the Dbx1 status confers interconnectivity between preBotz and VRG, then it could also account for interconnectivity between preBotz neurons to form a rhythmic network. Yet the VRG do not, as argued, form interconnections. Perhaps change "rostral group" to PreBotzinger neurons in the penultimate sentence. Or break this idea off into a separate paragraph with simpler wording.

4. It is interesting that both preBotz and VRG share a common transcription factor Dbx1. While this may account for PreBotz to VRG connections, it does not appear to account for VRG to phrenic MNs. So I'm not sure the statement, "... are sufficient for building an inspiratory motor circuit ..." is entirely accurate, since VRGs presumably need a mechanism to discriminate respiratory MNs from ambulatory subtypes, and from other spinal neuronal classes.

5. The final section of the results, describing the effects of Robo3 deletion on respiratory function is hard to follow. In particular, can the authors provide a clearer explanation (for both the L-R desynchronization and plethysmography) of how they discriminate effects of Robo loss on VRG versus preBotz circuits? I think this is important in light of the fact that the Robo3 mutation was described in a previous study by the authors. On p.10 the authors should introduce the effects of this mutation with greater clarity, state what they had previously shown in their analyses of the preBotz CPG, and explain how this analysis differs from the previous study.

Reviewer #3 (Remarks to the Author):

General comments.

Delineating the structural organization of brainstem neural circuits producing breathing movements in mammals is a fundamental problem of widespread interest in the motor systems neuroscience field. A major longstanding, unresolved problem addressed in the present study is to establish spatial organization and developmental identities of brainstem premotor neurons forming circuits that provide inspiratory motor drive to cervical spinal motoneurons (especially phrenic motoneurons) that produce coordinated bilateral activation of the diaphragm generating inspiratory movements of the respiratory pump. The authors have elegantly tackled this problem by employing in neonatal mice state-of-the-art transsynaptic rabies viral-based anatomical circuit mapping (pioneered by the authors) in combination

with sophisticated mouse transgenic strategies to delineate genetic lineage and locations of premotoneurons projecting to the bilateral phrenic motor nucleus. All of the technical aspects of the study, including the histochemical, electrophysiological, optogenetic, and behavioral analyses performed to verify aspects of the circuit organization/connectivity defined by the transsynaptic viral labeling and genetic manipulations are rigorously executed. In addition to establishing major features of the spatial organization of phrenic premotoneurons (Ph-preMNs), essential new information is provided on the developmental origin and neurotransmitter identities of these neurons.

An important major new finding is that the bulk of Ph-preMNs, confirmed here with novel and more informative tracing schemes to be located in the rVRG of the ventrolateral medulla, originate from transcription factor Dbx1-expressing ventral progenitors and have a V0, glutamatergic identity, while a smaller number of rVRG Ph-preMNs originate from Lbx1-expressing progenitors (dB identity) and are inhibitory (expressing GAD1 and GlyT2). Another important aspect is that the authors have established V0 homotypic connectivity from the preBötC region (where the inspiratory rhythm originates) to the rVRG, by a clever tracing scheme with a transgenic mouse line construct that allowed supplementary retrograde transsynaptic transport from rVRG V0 Ph-preMNs to upstream presynaptic preBötC V0 interneurons. This connectivity has been suspected, but not definitively established as the present study accomplishes.

Other important new results include: (1) some Ph-preMNs in the rVRG and preBötC (and other premotor populations delineated here) have bilaterally projecting branched axons onto the phrenic motor pool that likely contribute to bilaterally coordinated contraction of the hemi-diaphragms; (2) the surprising demonstration that rVRG V0 Ph-preMNs are not intrinsically connected, in contrast to what has been established for the upstream V0 preBötC rhythm generator population as further demonstrated here; (3) rVRG V0 Ph-preMNs do not send collateral projections to major respiratory-related cranial motoneurons; (4) the demonstration by optogenetic stimulation experiments that the rVRG glutamatergic premotoneurons have functionally intact bilateral synaptic connectivity to Ph-MNs at embryonic day E15.5 (when fetal breathing occurs); and (5) genetically deleting commissural axons leads to asynchronous preBötC V0 neuronal activity and unbalanced bilateral synchronous activity of phrenic motoneuronal electrophysiological outputs in vitro at E15.5, and also causes abnormal timing and bilateral amplitude synchronization of hemi-diaphragmatic activity with associated abnormal ventilatory patterns in P0 neonates in vivo.

The manuscript is well written and the principal results are very nicely illustrated in the main and supplementary figures. In general, the results presented are convincing, and support the important main conclusion that preBötC and rVRG Dbx1-derived V0 glutamatergic neurons form a core circuit that is synaptically organized to insure bilaterally synchronized inspiratory rhythmic activity and left-right balanced, synchronized premotor drives required for optimal operation of the respiratory pump to support survival at birth.

I have a few suggestions for revision of the manuscript.

Specific comments for manuscript revision.

Introduction

p. 4, para. 2, lines 1-3. "However, nothing is known of the identities of premotoneurons..." This statement is somewhat at odds with the authors' statement in Discussion (p. 14, para. 1): "Altogether, these locations are in agreement with previous anatomical and electrophysiological delineations of Ph-preMNs..." I suggest reformulating this statement to make clearer what is meant by "identities of premotoneurons and of the specific synapses..."

p. 4, para. 2, lines 14-15. "... rVRG neurons share with preBötC neurons, not only a glutamatergic and commissural phenotype..." I suggest qualifying here what is meant by "commissural phenotype" since this phenotype is different in detail.

Results

p. 5, para. 1, line 11. "...but in no other location." Please indicate in Methods, the extent to which suprabrainstem regions were surveyed.

Discussion

p. 13, para. 2, lines 5-6 & p. 14, line 1. "The bulk (about 85%) of brainstem Ph-preMNs locate to the ventral respiratory column in the BötC and, most prominently, in the rVRG." This statement does not correspond to the data presented in Supplementary Table 1. Please revise as required.

Methods

p. 25, para. 2. The authors appropriately indicate here caveats about variability in the numbers of premotoneurons labeled in a given region between animals, and potential quantitative biases related to variability in the number of seeding Ph-MNs. Another aspect to be discussed is the use of the G-deficient rabies virus with the AAV-G (serotype 6), which seems to improve the population labeling compared to that with the EnvA delta-G-RB on the ChATcre; R26ssHTB background, at least as revealed by Supplementary Fig. 1. Please comment on this aspect. In general, it would be appropriate to indicate the efficiency with which there is adequate coverage of the Ph-preMNs with the retrograde labeling strategies employed.

Supplementary Information

Supplementary Table 1. This table does not include the labeling of raphe neurons mentioned in the main text.

Reviewers' expertise:

Reviewer #1: Breathing circuits;

Reviewer #2: Development of motor circuits, neural circuit assembly during development;

Reviewer #3: Breathing circuits.

Reviewers' comments:

We should like to thank the reviewers for their positive and constructive comments on the manuscript. Due to the re-ordering of sections according to Nature Communication guidelines, the METHODS caption now follows the DISCUSSION. As a consequence, text changes that relate to your comments on the METHODS, now appear with about 4 page advancement with respect to where you had located your concerns.

Reviewer #1 (Remarks to the Author):

All behavior requires movement, with the obvious consequence that the critical structures in the nervous system producing the behavior must be connected to the relevant motoneurons. For breathing, the primary inspiratory muscle is the diaphragm, innervated by phrenic motoneurons. Since the inspiratory rhythm generator is the preBotzinger Complex (preBötZ) there must be a rather direct pathway between the two that is mostly represented by premotor neurons in the rostral ventral respiratory group (rVRG). Here the authors use a contemporary, highly sophisticated viral-based track tracing technique in mice to identify the location and phenotype of inspiratory premotor neurons and to delineate some of their properties and projections. The study is done with admirable care and precision, and the data is impeccable and valuable. The authors considerably advance previous studies by defining the transcription lineage of premotor neurons, conclusively establishing their transmitter phenotypes and the laterality of their projections, and show the importance of the V0 lineage in the connections between the preBötZ and rVRG. One important conclusion is that the majority of vGlut2+ Ph-preMNs resides in the rVRG where they appear to have an exclusive V0 identity. This and their other interpretations are fair and reasonable and represent a significant advance in understanding the motor control of inspiratory movements. Most of my comments related to the INTRODUCTION and DISCUSSION, which are mostly minor.

We thank the reviewer for his (her) comments on the manuscript.

SUMMARY

“These motor drives emerge from interactions between critical sets of brainstem neurons whose identities and synaptic ordered organization remain unresolved.” I don’t understand “unresolved”. Certainly, there are numerous published papers. What is “unresolved”?

Right, this sentence was intended to mean that the “identities” (in the sense of “developmental identities”) or more simply the “origins” of critical sets of

respiratory neurons and their connectivity are still obscure. This has now been corrected.

Text change: Page 2, line 4, “These motor drives emerge from interactions between critical sets of brainstem neurons whose origins and synaptic ordered organization remain obscure.”

“...the principal inspiratory premotor neurons share V0 identity with, and are connected by, neurons of the preBötzinger complex that pace inspiration.” While the preBötz paces inspiration, the preBötz to rVRG projections may not be from the neurons that actually “pace” inspiration.

We agree with the reviewer that there was indeed a grammatical error: the subject of the verb “to pace” is the preBötC and should therefore be properly conjugated as “paces”.

Text change: Page 2, line 8, “the principal inspiratory premotor neurons share V0 identity with, and are connected by, neurons of the preBötzinger complex that paces inspiration.”

“Deleting the commissural projections of V0s results in left-right desynchronized inspiratory motor commands in reduced brain preparations and breathing at birth.” Species?

Right, “in the mouse” has been inserted.

Text Change: Page 2, line 6, “Here, we show, using a virus-based transsynaptic tracing strategy from the diaphragm muscle in the mouse, that...”

INTRODUCTION

“In mammals, breathing is a motor behavior generated by a central pattern generator (CPG) located in the brainstem”. CPGs usually include motor neurons, so the respiratory CPG is in the brainstem AND spinal cord.

Thanks, we have re-phrased accordingly.

Text change: Page 3, 1st para, line 2, “In mammals, breathing is a motor behavior generated by a central pattern generator (CPG) located in the brainstem and spinal cord that produces rhythmic contraction of muscles that regulates lung volume and control upper airway patency to maintain bodily homeostasis ¹.”

“...allowing for breathing practice period prior to the challenge of encountering air at birth.” Typo.

The typo has been corrected.

Text Change: Page 3, 1st para, line 6, “... allowing a period of breathing practice prior to the challenge of encountering air at birth.”

“The CPG respects two intangible constraints, namely the synchronicity and the balanced amplitude...” balanced amplitude ??? Is this true in all cases for all inspiratory muscles?

The concern about “balanced amplitudes” unlike “synchronicity” of left and right drives to hemi-diaphragms is to our knowledge very poorly documented and we couldn’t find any direct reference and certainly no statement that this would be the case for all inspiratory muscles under any circumstances. This said together with alternating inspiratory/expiratory phases, amplitude balanced motor drives to left and right respiratory muscles appears adapted to the design of the upper airways that converge on a unique tract imposing unidirectional air flows, in or out (syringes work best with a single piston). Our work possibly provides the first animal model (*Dbx1^{cre};Robo3^{lox/lox}* mutants, admitted in E15.5 embryos) featuring mismatching amplitudes (but preserved synchronicity) of left and right motor drives to Ph-MNs strongly suggesting that wildtypes do elaborate left/right amplitude balanced drives. Careful examination of this point will probably need to re-investigate mechanical coupling of the hemi-diaphragms and the conflicting evidences that the different parts of the diaphragm contracting in situ act mainly in series or in parallel (Macklen et al., J. Appl. Physiol. 1983; Bellemare et al., 1986).

We have re-phrased to make our statement less assertive.

Text change: Page 3, 1st para, line 12, “The CPG must probably do so respecting two constraints, namely the synchronicity and the balanced amplitude of the motor drives onto left and right respiratory effector muscles (e.g. left and right costal diaphragm muscles that are the prime movers of tidal air).”

“The identity of neurons in charge of ensuring fail-safe bilaterally synchronized and amplitude balanced inspiratory motor drive is investigated here.” What about this work addresses “fail-safe”?

Our paper will make the demonstration that the core executive control circuit for inspiration is comprised of a V0 bilaterally synchronized rhythm generator feeding a V0 bilaterally projecting V0 rVRG premotor station that includes neurons with branched axons projecting both onto Ph-MNs on the left and right side, and is therefore optimally designed to “secure” synchronous mirror drives to left and right pools of Ph-MNs.

We agree that “ensuring fail-safe” is not very straightforward and in fact may be a definition for “secure”. Hence, we have replaced “ensuring fail-safe” by “securing”.

Text change: Page 3, 1st para, line 16, “The identity of neurons in charge of securing bilaterally synchronized and amplitude balanced inspiratory motor drive is investigated here.”

P4 “However, nothing is known of the identities of premotor neurons” Nothing? Isn't there considerable literature delineating the location, afferent and efferent

projections, transmitter phenotype, etc. of these neurons. In the DISCUSSION, the authors acknowledge this: “Altogether, these locations are in agreement with previous anatomical and electrophysiological delineations of Ph-preMNs made in the adult mouse 23, rat 24 and cat 25,” but without a similar statement earlier, the naïve reader may think “nothing” is known.

Thanks for raising this concern, to both avoid that the naïve reader be misled and to clarify our point we have reworded and simplified this sentence dropping the reference to “specific synapses”.

Text Change: Page 4, 2nd para, first line, “Although inspiratory descending circuits have been described for adult rodents and cats ⁸⁻¹⁰, nothing is known of the origin of premotor neurons downstream of the rhythm generator that secure temporal and amplitude patterning of the inspiratory motor drive.”

“and of the specific synapses downstream the rhythm generator that secure temporal and amplitude patterning of the inspiratory motor drive”, More precision is needed here. Certainly, we know that the preBötz projects to rVRG premotor neurons that in turn project to the diaphragm (see previous comment). The value added in this paper relates to further refinement of the neuronal phenotypes. Also typo “downstream OF...”

Thanks, this point has been dealt with above along with the typo.

“Here we addressed this question using viral-based circuit-mapping approaches 8 from the diaphragm muscle in early postnatal mice. We find that phrenic premotor neurons (Ph-preMNs) are distributed at several sites of the brainstem and include neurons with bifurcating axons that connect to phrenic motor neurons (Ph-MNs) on both sides of the midline.” Dobbins et al (REF 24) used “viral-based circuit-mapping” approaches in another rodent species (rat) and basically had the same results as described here, as the authors ultimately acknowledge in the DISCUSSION. The authors need to emphasize the novelty of their approach, and their new/novel/more refined findings.

This comment mostly concerns the first sentence of the para as the following ones allude to novel aspects of our work though in a succinct manner in this introductory section. To highlight the methodological originality of our work (a relapsing concern see also the next point) we have reworded this first sentence.

Text change: Page 4, 2nd para, line 6, “Here we addressed this question in early postnatal mice using monosynaptic viral-based circuit-mapping approaches ⁸ that allow unambiguous identification of phrenic premotor neurons (Ph-preMNs). We find that Ph-preMNs...”

RESULTS

P4 “To selectively label neurons that synapse onto Ph-MNs, we used transsynaptic rabies technology with monosynaptic restriction.” The actual technique used provides significantly more precision than prior use of

“transsynaptic rabies technology WITHOUT monosynaptic restriction” (REF 24). It would be useful for the authors to acknowledge this and point out what is novel here.

Thanks, the merits of this technique central to multiple insightful studies published in high profile journals since 2008 are now well accepted and are essentially recapitulated in the first sentence of the RESULTS by: “To selectively label neurons that synapse onto Ph-MNs...”

Furthermore, to take into account your remark about the providing “significantly more precision” we also have added a statement in the third sentence of the RESULTS:

Text change: Page 5, line 5, “Once inside presynaptic neurons, the deficient virus ceases to spread for lack of G, and thus only phrenic premotor neurons are traced safe of the confounds normally associated to multi-synaptic jumps of the non-deficient rabies virus.”

P5, bottom: How do these results compare to REF 24 in rats? This is in the DISCUSSION, but a comment here might be useful.

We don’t agree to repeatedly make reference to this particular study especially in the RESULTS where it can only disrupt the flow. We will (see below) agree to your point that “substantial conclusions” are best in the DISCUSSION, not in the RESULTS and consider that it all the more applies to this “comment”. Note that we also refrain from quoting these past studies in negative contexts like at the end of the sentence produced in response to the immediately preceding point.

P8 “Taken together, our data establish that the majority of vGlut2+ Ph-preMNs resides in the rVRG where they have an exclusive V0 identity.” Need to rephrase. This may be true since a similar proportion of trace+ neurons are VGlut2+ or Dbx1+ with a similar expression of Pax2, but no data showing that all VGlut2+ Ph-preMNs in the rVRG are Dbx1+. Rather, it would be fair to say that trace+ neurons of V0 origin appear exclusively in the rVRG (and preBötz).

We agree with the reviewer that although the statement is correct regarding residence of the majority of vGlut2+ Ph-preMNs in the rVRG, we cannot conclude to their exclusive V0 identity.

With this in mind, we take your point that after verifying that no other major Ph-preMN population has a V0 identity, we should rather conclude on the almost exclusive residence of the latter in the rVRG. We have therefore, re-phrased to meet both demands.

Text Change: Page 8, 2nd para, last sentence, “Therefore, these data establish that the trace+ glutamatergic Ph-preMNs with V0 identity reside almost exclusively in the rVRG.”

P9 “We first verified the massive presence of rVRG neuron synaptic terminals (7.7 ± 0.52 boutons/soma, $n=62$ cells) on motor neurons of the Ph-MN pool (Fig. 4a-c).” This was previously shown (without quantification or such

sophistication) by Feldman et al. J. Neurosci, 1985.

Not too sure what to make of this comment.

Here “first” relates to the result flow and is not intended to mean that authors are the first to...In order to make this clearer we have reformulated to: “We verified first....”

We do not consider that this reference should be prioritized given that the animal model, the tracer used there (tritiated amino acid in the cat) and the conclusions reached regarding the bilateral distribution of VRG projections, differ substantially from ours for reasons that may include the limited selectivity of the tracer entry towards rVRG cells (see Figure 1A and 2A in Feldman et al., 1985). Furthermore, the work by the group of Jack Feldman is already acknowledged by 6 citations in this ms.

Text change: Page 9, 1st para, line 4, “We verified first the massive presence of rVRG neuron synaptic terminals (7.7 ± 0.52 boutons/soma, $n=62$ cells) on motor neurons of the Ph-MN pool (Fig. 4a-c).”

P9 “Together these data indicate that rVRG neurons are not intrinsically connected and that they send collateral projections to the LRN but not to major respiratory-related cranial motor neurons.” Need to rephrase. Fig. 4d-f shows that V0 trace+ rVRG neurons do not synapse onto the somas of other trace+ rVRG neurons, not that the rVRG lacks interconnectivity or only projects to the LRN.

The reviewer is right that we should refer to trace+ rVRG neurons not to the rVRG as a whole. We have re-written this paragraph making systematic references (6 replacements) to “trace+ rVRG neurons”, including in the final statement below.

Text change: Page 9, 1st para, line 20, “Together these data indicate that trace[±] rVRG neurons are not intrinsically connected and that they send collateral projections to the LRN but not to major respiratory-related cranial motor neurons.”

Also in Fig. 4g there appears be yellow V0 trace+ puncta in the projection field dorsal to the LRN.

This is a wrong impression owing to the presence of red labeled processes in the low magnification z-projections in Fig.4 g and h. For the reviewer’s eyes only we have produced an illustration (below) of the projection field dorsal to the LRN (g’, h’) which shows the virtual absence of double labeled puncta in a single optical section (i’) in agreement with our statement that “Apart from Ph-MNs, the only conspicuous presence of trace+ rVRG synaptic terminals was in the ipsi- and contra-lateral lateral reticular nucleus (LRN)....”

Could there be synapses onto trace- rVRG neurons?

This is very unlikely as yellow puncta are not conspicuous in the rVRG region and are virtually absent from trace⁺ rVRG neurons, furthermore, the status (premotor or not) of trace⁻ rVRG neurons in the vicinity of trace⁺ rVRG neurons cannot be ascertained.

P12 “We conclude that two sets of Dbx1-derived excitatory neurons redundantly ensure (i) generation of a bilaterally synchronized inspiratory rhythm and (ii) the ... synchronized premotor drives required for optimal efficiency of the respiratory pump at birth.” Substantial conclusions are best in the DISCUSSION, not in the RESULTS.

We have as suggested, removed this conclusion from the RESULTS.

DISCUSSION

P 13 “Using trans-synaptic tracing strategies we have now revealed the yet uncharted identities of a fourth essential component: phrenic premotor neurons.” As noted above, the authors reveal novel aspects (is this what “uncharted identities” mean?) of this projection, but give the impression that almost “nothing” was known prior to this work (as so stated earlier by the authors, commented above).

This point has been addressed previously and we have re-worded here accordingly.

Text change: Page 13, 2nd para, line 11, “Using trans-synaptic tracing strategies we have now revealed the yet uncharted origins of a fourth essential component: phrenic premotor neurons.”

P13-14 “excitatory vGlut2+ Ph-preMNs have a P0 origin and almost exclusively reside in the rVRG.” Typo “V0”, and rephrase since not shown that all vGlut2+ Ph-preMNs are Dbx1+.

Strictly speaking p0 is not a typo, neurons originate in progenitors that give rise upon cell cycle exit to neurons that can be given neuronal type or subtype

identities. Progenitors that express Dbx1 are known as p0 progenitors that will give rise to neurons with V0 type identity. Thus neurons cannot have a V0 origin. To deal here with your point on vGlut2/V0 relationship we have re-worded our sentence.

Text Change: Page 15, 2nd para, line 9, “Inhibitory vGAT+ Ph-preMNs have a dB1/dB4 origin, form all of the BötC and contribute a third of rVRG neurons, while excitatory vGlut2+ Ph-preMNs by large of p0 origin reside in the rVRG.”

P15 “One major finding is that the predominant premotor group, the glutamatergic rVRG neurons have a V0 identity..” Same as above

In line with your preceding point, we have also reworded here.

Text change: Page 15, last para, line 2, “One major finding is that the predominant premotor group, the glutamatergic rVRG neurons have for the most part a V0 identity, like the preBötC.”

P 16 “A novel trait, not revealed by traditional tracing methods, is that Ph-preMNs display a special axonal morphology: a bilaterally branched axon projecting onto phrenic motor pools on both sides of the midline.” This was previously shown in other species, so the novelty is the identification in mice not the finding itself.

We are inclined to consider that the only undisputable way to identify this axonal morphology is through the use of double retrograde trans-synaptic labeling as performed here and in the mentioned previous works in refs 35 and 36. We are not aware of preceding reports in other species in which this technique was used. In fact, previous reports only now stand a chance to be validated by confrontation with results obtained by novel trans-synaptic tracing schemes. As it turns out our data are largely in agreement with the early work, this should be considered by readers (including past authors) reassuring, not trivial.

To make our sentence best descriptive of the specificity of our work and to eliminate the present criticism we have reworded our sentence by replacing “projecting” by “synapsing”.

Text change: Page 16, 2nd para, line 3, “A novel trait, not revealed by traditional tracing methods, is that Ph-preMNs display a special axonal morphology: a bilaterally branched axon synapsing onto phrenic motor pools on both sides of the midline.”

SUPPLEMENTAL FIGURES

SUPP Fig. 1e Trace+ neurons appear to be in the supratrigeminal nucleus rather than the PB/KF- consider using another representative image.

Thanks, we have now replaced the image of the Supplementary Figure 1e panel to make it more representative of the PB/KF location.

Reviewer #2 (Remarks to the Author):

In the study by Wu and colleagues the authors investigated the developmental origin and anatomical organization of neural populations and circuits that control breathing. Using rabies virus-mediated transsynaptic labeling on genetically defined populations of brainstem neurons, the authors characterize the neuronal subtypes targeting phrenic motor neurons. The authors show that the predominant population of premotor neurons targeting phrenic motor neurons are located in the rostral ventral respiratory group. Similar to the rhythmically active neurons in preBotzinger complex, rVRG neurons are derived from interneurons that express the transcription factor Dbx1. They show that these rVRG neurons are glutamatergic, and VRG neurons target both ipsilateral and contralateral MN populations. Using Robo3 conditional mutants to disrupt the commissural projections of V0 neurons, the authors characterize the function of VRG neurons in coordinating bilateral muscle contraction.

Previous studies by the authors and others have characterized the developmental origin of the central pattern generating networks in the brainstem, as well as the effector motor neuron subtypes that drive respiratory muscle. How brainstem networks engage respiratory motor neurons is less clear, and this study provides an important link between these two systems. The authors make the interesting observation that both preBotz and VRG neurons derive from populations expressing the Dbx1 transcription factor.

Overall this is an excellent paper, and provides a major contribution to our understanding of the basic architecture of the neural circuits that control breathing. The study beautifully characterizes anatomical organization of prePhrenic MNs. The results are easy to follow and the data are well quantified. The paper generally well written, with the exception of the last two data figures, which need minor revision (see below).

We thank the reviewer for his (her) comments on the manuscript.

Major comments.

Thank you, we realize from your first four points, that our paragraph linking the present data to a speculative evolutionary scenario of the motorization of the diaphragm was difficult to penetrate. This para has now been entirely re-written taking on board your concerns and remarks.

1. In addition to phrenic motor neurons, respiratory function requires coordination of intercostal and abdominal muscle. Presumably in amniotes which lack a diaphragm (such as in birds and reptiles) axial muscles provide the major inspiratory drive. Can the authors comment on this?

There is here no room in the present ms for commenting on this particular point. [For your info, there is indeed a consensus that aspiration breathing involves almost exclusively body wall muscles and their derivatives that once formed the locomotor apparatus in fish. In fact, no amniotes have evolved complete mechanical separation of locomotor and respiratory functions of body wall

muscles. There are fascinating studies in lizard species in which the same muscles must be activated in different sequences to allow locomotion or breathing resulting in animals that cannot breathe and walk at the same time. This phenomenon, known as axial constraint (Carrier DR, Amer Zool 1991) limiting costal breathing can be compensated by supplemental buccal pump ventilation in some species (Al Ghamdi et al., J Exp Biol 2001) and is overcome in mammals by use of a muscular septum –the diaphragm- for inspiration.]

We do make it clear that motorization of the diaphragm required that “respiratory activity was shifted from branchiomotor nerves to spinal somatic nerves (e.g. the phrenic nerve)...”

For example, do rVRG neurons also target other MN subtypes in addition to the phrenic as shown in Figure 4a-c?

In the paper we explicitly report that we failed to identify trace+ rVRG neurons synapses (identified by Dbx1synGFP+/and mCherry+) onto cranial motoneurons. We have unpublished data showing that tracing from intercostal muscles also transsynaptically labels rVRG neurons but at present we have not yet used distinctly labeled viral vectors in the diaphragm and intercostal muscles to identify individual double labeled rVRG neurons that would project to both motoneuronal sets. Obviously, comparative physiological and anatomical investigations of central circuits controlling ventilation in birds and reptiles need be performed even if only to address the presence there of a preBötC-like rhythm generator bearing V0 identity.

2. Also the evolutionary arguments in the discussion imply the breathing circuits were initially for “Motorizing the diaphragm”. But the diaphragm has only been described in mammals, so it seems that there were at least two independent events (first a PreBotz-VRG-axial respiratory circuit; then later a VRG-phrenic). I am unsure how well this has been studied in chick or other model system, but it seems the evolutionary history is more complex than stated.

There is a misunderstanding here that we hope the new para will help clarify. We understand that the previous use of “...motorizing the diaphragm in the first amniotes...” may have been taken to suggest erroneously that the first amniotes already had evolved a diaphragm while we intended to mean “the first amniotes” i.e. those that had first evolved a diaphragm. We now have found a more direct way to express this.

Test change: Page 18, 2nd para, line3, “The amniotes that first evolved a diaphragm required ...”

3. I would also suggest the authors consider restating the argument in the last paragraphs of the discussion again for clarity. I think the hypothesis about two distinct steps in V0 diversification is an interesting hypothesis, but the way it is written is hard to follow. If the Dbx1 status confers interconnectivity between preBotz and VRG, then it could also account for interconnectivity between preBotz neurons to form a rhythmic network. Yet the VRG do not, as argued,

form interconnections. Perhaps change “rostral group” to PreBotzinger neurons in the penultimate sentence. Or break this idea off into a separate paragraph with simpler wording.

Thank you we have indeed reworded to ease the logical flow and have made clear that the rostral group qualifies the preBötC as suggested in the incriminated sentence.

Text change: Page 18, 2nd para, last sentence, “We hypothesize that during evolution two neighboring groups of hindbrain commissural premotor V0 neurons simultaneously acquired a glutamatergic phenotype while the rostral group only, the preBötC (possibly through a rhombomere specific-mechanism), acquired the capacity to connect, rather than to motor neurons, to other V0 neurons, i.e. to form homotypic V0-V0 synapses.”

4. It is interesting that both preBotz and VRG share a common transcription factor Dbx1. While this may account for PreBotz to VRG connections, it does not appear to account for VRG to phrenic MNs. So I’m not sure the statement, “... are sufficient for building an inspiratory motor circuit ...” is entirely accurate, since VRGs presumably need a mechanisms to discriminate respiratory MNs from ambulatory subtypes, and from other spinal neuronal classes.

Thank you, we have realized that the previous version of this discussion paragraph did not read easily. We are not in a position to discuss, let alone provide insights on, the multiple changes that contributed to the emergence of aspiration breathing in tetrapods. We hope that the present version makes it clear that assuming a default premotor somatic fate for V0 neurons, admitted leaving aside the issue of motor neuronal target recognition, allows given the data here produced, to speculate on V0 fate changes that may bear relevance to the appearance of a central circuit controlling inspiration.

Text change: Page 18, 2nd para, line 7, “In the spinal cord, V0 neurons were first described as commissural inhibitory premotor neurons synapsing onto somatic motor neurons that innervate hindlimb muscles⁴⁸. Assuming that this is the default state of V0s produced throughout the neuroaxis...”

5. The final section of the results, describing the effects of Robo3 deletion on respiratory function is hard to follow. In particular, can the authors provide a clearer explanation (for both the L-R desynchronization and plethysmography) of how they discriminate effects of Robo loss on VRG versus preBotz circuits? I think this is important in light of the fact that the Robo3 mutation was described in a previous study by the authors. On p.10 the authors should introduce the effects of this mutation with greater clarity, state what they had previously shown in their analyses of the preBotz CPG, and explain how this analysis differs from the previous study.

Thank you, your comment calls for introducing statements in the text to explicitly mention (i) that the *Dbx1^{cre};Robo3^{lox/lox}* disrupts the commissural navigation of all Dbx1-derived neurons i.e. including those of the preBötC and of the rVRG, (ii) the effect of the Robo3 mutation, (iii) previous results and what had not been done previously. We therefore have introduced three novel

sentences to deal with these issues.

Text change: Page 11, line1, “Second, we prevented the commissural navigation of the axons of V0 neurons by deleting the Robo3 gene with a Dbx1cre line ⁷. In the absence of Robo3 receptor-mediated signaling, axons fail to navigate across the midline²⁰. This conditional interference collectively targets V0 neurons of the preBötC causing left-right de-synchronisation of its activity ⁷ and those of the rVRG whose role in this context had not been previously investigated. The impairment of the rVRG...”

Reviewer #3 (Remarks to the Author):

General comments.

Delineating the structural organization of brainstem neural circuits producing breathing movements in mammals is a fundamental problem of widespread interest in the motor systems neuroscience field. A major longstanding, unresolved problem addressed in the present study is to establish spatial organization and developmental identities of brainstem premotor neurons forming circuits that provide inspiratory motor drive to cervical spinal motoneurons (especially phrenic motoneurons) that produce coordinated bilateral activation of the diaphragm generating inspiratory movements of the respiratory pump. The authors have elegantly tackled this problem by employing in neonatal mice state-of-the-art transsynaptic rabies viral-based anatomical circuit mapping (pioneered by the authors) in combination with sophisticated mouse transgenic strategies to delineate genetic lineage and locations of premotoneurons projecting to the bilateral phrenic motor nucleus. All of the technical aspects of the study, including the histochemical, electrophysiological, optogenetic, and behavioral analyses performed to verify aspects of the circuit organization/connectivity defined by the transsynaptic viral labeling and genetic manipulations are rigorously executed. In addition to establishing major features of the spatial organization of phrenic premotoneurons (Ph-preMNs), essential new information is provided on the developmental origin and neurotransmitter identities of these neurons.

An important major new finding is that the bulk of Ph-preMNs, confirmed here with novel and more informative tracing schemes to be located in the rVRG of the ventrolateral medulla, originate from transcription factor Dbx1-expressing ventral progenitors and have a V0, glutamatergic identity, while a smaller number of rVRG Ph-preMNs originate from Lbx1-expressing progenitors (dB identity) and are inhibitory (expressing GAD1 and GlyT2). Another important aspect is that the authors have established V0 homotypic connectivity from the preBötC region (where the inspiratory rhythm originates) to the rVRG, by a clever tracing scheme with a transgenic mouse line construct that allowed supplementary retrograde transsynaptic transport from rVRG V0 Ph-preMNs to upstream presynaptic preBötC V0 interneurons. This connectivity has been suspected, but not definitively established as the present study accomplishes.

Other important new results include: (1) some Ph-preMNS in the rVRG and preBötC (and other premotor populations delineated here) have bilaterally projecting branched axons onto the phrenic motor pool that likely contribute to bilaterally coordinated contraction of the hemi-diaphragms; (2) the surprising demonstration that rVRG V0 Ph-preMNs are not intrinsically connected, in contrast to what has been established for the upstream V0 preBötC rhythm generator population as further demonstrated here; (3) rVRG V0 Ph-preMNs do not send collateral projections to major respiratory-related cranial motoneurons; (4) the demonstration by optogenetic stimulation experiments that the rVRG glutamatergic premotoneurons have functionally intact bilateral synaptic connectivity to Ph-MNs at embryonic day E15.5 (when fetal breathing occurs); and (5) genetically deleting commissural axons leads to asynchronous preBötC V0 neuronal activity and unbalanced bilateral synchronous activity of phrenic motoneuronal electrophysiological outputs in vitro at E15.5, and also causes abnormal timing and bilateral amplitude synchronization of hemi-diaphragmatic activity with associated abnormal ventilatory patterns in P0 neonates in vivo.

The manuscript is well written and the principal results are very nicely illustrated in the main and supplementary figures. In general, the results presented are convincing, and support the important main conclusion that preBötC and rVRG Dbx1-derived V0 glutamatergic neurons form a core circuit that is synaptically organized to insure bilaterally synchronized inspiratory rhythmic activity and left-right balanced, synchronized premotor drives required for optimal operation of the respiratory pump to support survival at birth.

We thank the reviewer for his (her) comments on the manuscript.

I have a few suggestions for revision of the manuscript.

Specific comments for manuscript revision.

Introduction

p. 4, para. 2, lines 1-3. “However, nothing is known of the identities of premotoneurons...” This statement is somewhat at odds with the authors’ statement in Discussion (p. 14, para. 1): “Altogether, these locations are in agreement with previous anatomical and electrophysiological delineations of Ph-preMNs...” I suggest reformulating this statement to make clearer what is meant by “identities of premotoneurons and of the specific synapses...”

Right, this point was also made by reviewer 1.

We take it that “identity” may indeed be interpreted in multiple ways while we intended to mean “developmental identity”. So we have reworded this sentence using the more accurate word “origin” and have chosen to make reference to previous work here for the “naïve” readers as suggested by reviewer 1. We have also dropped consideration about “the specific synapses” at this point.

Text change: Page 4, 2nd para, line 1, “Although inspiratory descending circuits

have been described for adult rodents and cats⁸⁻¹⁰, nothing is known of the origin of premotor neurons downstream of the rhythm generator that secure temporal and amplitude patterning of the inspiratory motor drive.”

p. 4, para. 2, lines 14-15. “... rVRG neurons share with preBötC neurons, not only a glutamatergic and commissural phenotype...” I suggest qualifying here what is meant by “commissural phenotype” since this phenotype is different in detail.

Thank you, for clarity we have reworded more simply this sentence.

Text change: Page 4, 2nd para, last sentence, “Strikingly, rVRG and preBötC neurons found both glutamatergic and harboring commissural axons share a common origin in p0 progenitors, highlighting the centrality of Dbx1-expressing neural progenitors in the advent of aspiration breathing in vertebrates.”

Results

p. 5, para. 1, line 11. “...but in no other location.” Please indicate in Methods, the extent to which suprabrainstem regions were surveyed.

We now explicitly indicate in Methods, that our survey considered the entire brain.

Text change: Page 21, 3rd para, line 1, “The virally infected premotor neurons were carefully surveyed in the entire brain rostral to the spinal cord.”

Discussion

p. 13, para. 2, lines 5-6 & p. 14, line 1. “The bulk (about 85%) of brainstem Ph-preMNs locate to the ventral respiratory column in the BötC and, most prominently, in the rVRG.” This statement does not correspond to the data presented in Supplementary Table 1. Please revise as required.

The reviewer is right this is a mistake that has now been corrected.

Text change: Page 14, 2nd para, line 6, “The bulk (about 75%) of brainstem Ph-preMNs locate...”

Methods

p. 25, para. 2. The authors appropriately indicate here caveats about variability in the numbers of premotoneurons labeled in a given region between animals, and potential quantitative biases related to variability in the number of seeding Ph-MNs. Another aspect to be discussed is the use of the G-deficient rabies virus with the AAV-G (serotype 6), which seems to improve the population labeling compared to that with the EnvA delta-G-RB on the ChATcre; R26ssHTB background, at least as revealed by Supplementary Fig. 1. Please comment on this aspect. In general, it would be appropriate to indicate the efficiency with which there is adequate coverage of the Ph-preMNs with the retrograde labeling strategies employed.

Thank you, the reviewer is right that the two tracing schemes that differ by neuronal targeting and means to achieve G-complementation (scheme 1: delta-G-Rb + AAV6-G) and (scheme 2: EnvA-delta-G-Rb + ChATcre;R26ssHTB) show

distinct efficiency. The latter one was found less effective and was only used to support the results obtained with the former and to rule out anterograde transsynaptic spread from putative diaphragm primary sensory afferents. Sub-optimal labeling is probably due to a reduced number of seeding Ph-MNs whose infection by an EnvA-pseudotyped Rb viral vector is conditioned by TVA expression from a cre-dependent transgene (R26ssHTB). We have incorporated two sentences just to indicate this point in the Methods section.

-Text change: Page 20, 1st para, last sentence, “The latter tracing scheme was found less effective than the original one. The total number of traced Ph-preMNs (counted in n=2 experiments) was found reduced by about 35% probably owing to suboptimal viral infection of seeding Ph-MNs in relation to limited cre-dependent expression of TVA from the *R26ssHTB* allele.”

Regarding efficiency we have added one statement in the Methods section.

-Text change: Page 22, 1st para, line 2, “Experiments that resulted in trans-synaptic labeling of less than 150 rVRG neurons (n=3) failed to label neurons in any other premotor location and were discarded.”

Supplementary Information

Supplementary Table 1. This table does not include the labeling of raphe neurons mentioned in the main text.

We had excluded raphe neurons from the table that quantifies the bilateral distribution of Ph-preMNs because the midline location of their soma prevented unambiguous distinction of their ipsi- or contra-lateral position. We have now added the quantification of raphe neurons to Supplementary Table 1 with no ipsi- vs contra-lateral distinction.

REVIEWERS' COMMENTS:

Reviewer #1 (Remarks to the Author):

The authors response to all 3 reviewers was respectful, meticulous and on point. The resulting reacted manuscript improves on an already excellent and (generally) interesting paper.

Reviewer #2 (Remarks to the Author):

I am very satisfied with the authors response to my and the other reviewers comments. They did a very thorough job addressing all the issues with the paper, most of which were textual. I think this paper will be an important contribution to the field.

Reviewer #3 (Remarks to the Author):

The authors have satisfactorily addressed my suggestions for revision of the manuscript, which is now improved. The novel results and conclusions presented, as outlined in my original review, are further strengthened in this revision. This manuscript represents a major contribution that advances our understanding of the spatial organization and developmental identities of brainstem premotor neurons forming circuits providing inspiratory motor drive to phrenic motoneurons generating inspiratory movements of the respiratory pump in mammals.